# OVER-PARAMETERIZED MODEL OPTIMIZATION WITH POLYAK-ŁOJASIEWICZ CONDITION

**Yixuan Chen[1], Yubin Shi[1], Mingzhi Dong[1,*], Xiaochen Yang[2,*], Dongsheng Li[3],**
**Yujiang Wang[4], Robert P. Dick[5], Qin Lv[6], Yingying Zhao[1], Fan Yang[7], Ning Gu[1], Li Shang[1]**
`{yixuanchen20,ybshi21}@fudan.edu.cn`

## ABSTRACT

This work pursues the optimization of over-parameterized deep models for superior training efficiency and test performance. We first theoretically emphasize the importance of two properties of over-parameterized models, i.e., the convergence gap and the generalization gap. Subsequent analyses unveil that these two gaps can be upper-bounded by the ratio of the Lipschitz constant and the Polyak-Łojasiewicz (PL) constant, a crucial term abbreviated as the *condition number*. Such discoveries have led to a structured pruning method with a novel pruning criterion. That is, we devise a gating network that dynamically detects and masks out those poorly-behaved nodes of a deep model during the training session. To this end, this gating network is learned via minimizing the *condition number* of the target model, and this process can be implemented as an extra regularization loss term. Experimental studies demonstrate that the proposed method outperforms the baselines in terms of both training efficiency and test performance, exhibiting the potential of generalizing to a variety of deep network architectures and tasks.

## 1 INTRODUCTION

Most practical deep models are over-parameterized with the model size exceeding the training sample size and can perfectly fit all training points (Du et al., 2018; Vaswani et al., 2019). Recent empirical and theoretical studies demonstrate that over-parameterization plays an essential role in model optimization and generalization (Liu et al., 2021b; Allen-Zhu et al., 2019). Indeed, a plethora of state-of-the-art models that are prevalent in the community are over-parameterized, such as Transformer-based models for natural language modeling tasks (Brown et al., 2020; Devlin et al., 2018; Liu et al., 2019) and wide residual networks for computer vision tasks (Zagoruyko & Komodakis, 2016). However, training over-parameterized models is usually time-consuming and can take anywhere from hours to weeks to complete. Notwithstanding some prior works (Liu et al., 2022; Belkin, 2021) on theoretical analyses of the over-parameterized models, those findings remain siloed from the common practices of training those networks.

The work seeks to optimize over-parameterized models, in pursuit of superior training efficiency and generalization capability. We first analyze two key theoretical properties of over-parameterized models, namely the convergence gap and the generalization gap, which can be quantified by the convergence rate and the sample complexity, respectively. Theoretical analysis of over-parameterized models is intrinsically challenging as the over-parameterized optimization landscape is often nonconvex, limiting convexity-based analysis. Inspired by recent research on the convergence analysis of neural networks and other non-linear systems (Bassily et al., 2018; Gupta et al., 2021; Oymak & Soltanolkotabi, 2019), we propose to use the Polyak-Łojasiewicz (PL) condition (Polyak, 1963; Karimi et al., 2016; Liu et al., 2022) as the primary mathematical tool to analyze convergence rate and sample complexity for over-parameterized models, along with the widely used Lipschitz con-

---

[1]China and Shanghai Key Laboratory of Data Science, School of Computer Science, Fudan University, Shanghai, China. [2]School of Mathematics Statistics, The University of Glasgow, Glasgow, UK. [3]Microsoft Research Asia, Shanghai, China. [4]Department of Engineering Science, University of Oxford, Oxford, England. [5]Department of Electrical Engineering and Computer Science, University of Michigan, Michigan, United States. [6]Department of Computer Science, University of Colorado Boulder, Boulder, Colorado, United States. [7]School of Microelectronics, Fudan University, Shanghai, China. [*]The corresponding author.

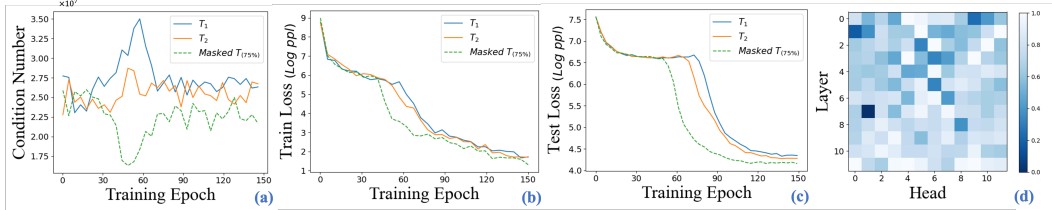

Figure 1: Benefits of PL regularization for BERT optimization. Figure (a-c) shows that models with a smaller condition number (e.g., $T_2 < T_1$ in general) achieve faster training convergence and better test performance. In addition, pruning heads with a large condition number, i.e., Masked $T$, reduces the condition number, leading to more rapid and accurate training. Figure (d) shows $T_2$ heads have different condition numbers. The largest ones are pruned to produce Masked $T$ in Figure (a-c).

dition (Allen-Zhu et al., 2019). *Our theoretical analysis shows that the aforementioned properties can be controlled by the ratio of the Lipschitz constant to the PL constant*, which is referred to as the condition number (Gupta et al., 2021). A small condition number indicates a large decrease in training loss after parameter updates and high algorithmic stability relative to data perturbation, i.e., fast convergence and good generalization ability. More promisingly, such pattern can be observed in empirical studies. As shown in Figure 1(a-c), where BERT models were applied to WikiText-2 for language modeling, the training loss of the model with a small condition number ($T_2$) decreases much faster than the model with a large condition number ($T_1$), especially when the differences in condition numbers are pronounced (between 40 and 80 epochs); its test performance also improves much faster and is ultimately better. Such theoretical and empirical findings motivate us to formulate a novel regularized optimization problem which adds the minimization of condition number to the objective function; we call this new additional term *PL regularization*. In this way, we can directly regularize the condition number while training over-parameterized models, thereby improving their convergence speed and generalization performance.

Our empirical analysis further reveals that, given an over-parameterized model, different model components exhibit distinct contributions to model optimization. Figure 1(d) plots the heatmap of the condition number for all model heads at epoch-10 and it shows that the condition number varies considerably between model heads. Given the fact that over-parameterized models contain a large number of redundant parameters, we argue that it is possible to reduce the condition number of an over-parameterized network during training by identifying and masking out poorly-behaved sub-networks with large condition numbers. Figure 1(a-c) illustrates the potential efficacy of this approach. After disabling 75% heads of the BERT model $T_2$ according to the condition number ranking at epoch-10, the masked BERT (*Masked T*) possesses a smaller condition number and achieves faster convergence and better test performance. This phenomenon motivates us to impose PL regularization, and hence improve model optimization, by adopting a pruning approach. More specifically, we introduce a binary mask for periodically sparsifying parameters, and the mask is learned via a gating network whose input summarizes the optimization dynamics of sub-networks in terms of PL regularization. An overview of the proposed method is provided in Appendix E.1.

The proposed pruning approach to enforcing PL regularization is related to the structured pruning works, which focuses on compressing model size while maintaining model accuracy. The significant difference lies in that we utilize the condition number to identify the important components, which thus can simultaneously guarantee the convergence and generalization of model training. More importantly, as a consequence of this difference, compared with most pruning works which obtain a sparse model at a slight cost of degraded accuracy, our method is found to achieve even better test performance than the dense model when no more than 75% of the parameters were pruned. Experimental results demonstrate that our method outperforms state-of-the-art pruning methods in terms of training efficiency and test performance. The contributions of this work are threefold:

- We are the first to propose using PL regularization in the training objective function of over-parameterized models. Such proposal is founded on the theoretical analysis of optimization and generalization properties of over-parameterized models, which shows that a small condition number implies fast convergence and good generalization.

- We describe a PL regularization-driven structured pruning method for over-parameterized model optimization. Specifically, we introduce a learnable mask, guided by the PL-based condition number, to dynamically sparsify poorly-behaved sub-networks during model training to optimize training efficiency and test performance.

- The proposed analysis framework and optimization method can be applied to a wide range of deep models and tasks. Experimental studies using three widely used deep models, namely BERT , Switch-Transformer , and VGG-16 , demonstrate that the proposed method outperforms the baselines on both training efficiency and test performance.

## 2 ANALYSIS OF OVER-PARAMETERIZED MODELS WITH PL CONDITION

In this section, we first show that, in the context of over-parameterized models, the optimal test error can be decomposed into convergence gap and generalization gap, which can be quantified via convergence rate and sample complexity, respectively. Next, using the tool of PL$^*$ condition, we show that both convergence rate and sample complexity are upper-bounded by the condition number and thus can be improved by using a small condition number.

### 2.1 OPTIMIZATION PROPERTIES OF OVER-PARAMETERIZED MODELS

Let $\mathcal{X}$ denote the input space and $\mathcal{Y}$ the target space. The standard optimization problem can be formulated as finding a minimizer of the expected risk: $f^* = \mathrm{argmin}_{f:f \in \mathcal{Y}^{\mathcal{X}}} R(f)$, where the expected risk is defined as $R(f) := \mathbb{E}_{(\mathbf{x},\mathbf{y}) \sim \mathcal{D}}[\mathcal{L}(f(\mathbf{x}), \mathbf{y})]$, $\mathcal{D}$ denotes an unknown data distribution over $\mathcal{X} \times \mathcal{Y}$, and $\mathcal{L}$ denotes a non-negative loss function. To restrict the search space, $f$ is typically chosen from a hypothesis class $\mathcal{H} \subset \mathcal{Y}^{\mathcal{X}}$; the minimizer in this case is given by $f_{\mathcal{H}}^* = \mathrm{argmin}_{f:f \in \mathcal{H}} R(f)$. In practice, the data distribution is unknown and thus the minimizer is found by minimizing the empirical risk: $f_{\mathcal{H},S}^* = \mathrm{argmin}_{f:f \in \mathcal{H}} R_S(f)$, where the empirical risk is defined as $R_S(f) := \frac{1}{n} \sum_{i=1}^{n} \mathcal{L}(f(\mathbf{x}_i), \mathbf{y}_i)$, and $S$ denotes the training set $S = \{(\mathbf{x}_i, \mathbf{y}_i)\}_{i=1}^{n}$ drawn i.i.d. from $\mathcal{D}$. The empirical risk minimization problem is often solved by using algorithms such as gradient descent; the optimal solution found by these algorithms is denoted by $\hat{f}_{\mathcal{H},S}$. Finally, when $f$ is parameterized by $\mathbf{w} \in \mathbb{R}^m$, we denote the loss function by $\mathcal{L}(\mathbf{x}, \mathbf{y}; \mathbf{w})$, or simply $\mathcal{L}(\mathbf{w})$, and also use $\mathcal{L}_S(\mathbf{w})$ to denote the empirical risk.

**Definition 1** *(Over-parameterized model Liu et al. (2020a)). A model is an over-parameterized model if the number of parameters $m$ is larger than the number of data.*

Motivated by the previous work (Gühring et al., 2020), the Bayes error, i.e., the optimal test error, can be decomposed into five components: empirical error, convergence gap, generalization gap, estimation error and approximation error. In the over-parameterized setting, following previous works (Brutzkus et al., 2017; Belkin, 2021; Liu et al., 2022), we assume that the optimal function $f_{\mathcal{H},S}^*$ fits or interpolates the training data exactly. Therefore, for over-parameterized models, the Bayes error decomposition can be simplified as

$$R(f^*) \leq 2|R_S(\hat{f}_{\mathcal{H},S}) - R_S(f_{\mathcal{H},S}^*)| + 3 \sup_{f \in \mathcal{H}} |R(f) - R_S(f)|. \tag{1}$$

The first and second terms refer to the convergence gap and the generalization gap, respectively. (The detailed decomposition can be found in Appendix A.) Thus, the Bayes error can be better approximated if we restrict the convergence gap and generalization gap. Furthermore, the convergence gap can be quantified in proportion to the reciprocal of **convergence rate**, and the number of instances required by specific generalization gap can be quantified in proportion to **sample complexity**.

**Definition 2** *(Convergence Rate (Alzubi et al., 2018)). If the model sequences $f(\mathbf{w}_0), \ldots, f(\mathbf{w}_t)$ during optimization converge to the optimal solution $f(\mathbf{w}^*)$, convergence rate can be defined as:*

$$\lim_{t \to \infty} \frac{f(\mathbf{w}_{t+1}) - f(\mathbf{w}^*)}{f(\mathbf{w}_t) - f(\mathbf{w}^*)} = \gamma, \tag{2}$$

where $\gamma$ is a real number in range $[0, 1]$. The convergence rate is measured by the limit of the ratio of successive differences. The sequence converges sublinearly, linearly, or superlinearly depending on whether $\gamma = 1$, $\gamma \in (0, 1)$, or $\gamma = 0$, respectively. For an algorithm with a linear rate of convergence, $R_S(f(\mathbf{w}_t)) \leq \exp(-t\gamma) (R_S(f(\mathbf{w}_0)))$.

**Definition 3** *(Sample Complexity (Alzubi et al., 2018)). For any $\varepsilon, \delta \in (0, 1)$, the sample complexity $n(\varepsilon, \delta)$ is defined as the smallest $n < \infty$ for which there exists a learning algorithm $f$ such that, for any distribution $\mathcal{D}$ on training dataset $S$ with the data sample $\mathbf{x}_1, \ldots, \mathbf{x}_n$ of size $n$,*

$$\Pr\left(|R(f_S) - R_S(f_S)| \leq \epsilon\right) \geq 1 - \delta. \tag{3}$$

The sample complexity denotes the number of required examples to guarantee a generalization gap smaller than $\epsilon$ with a probability $1 - \delta$.

## 2.2 PL* CONDITION AND THEORETICAL PROPERTIES

The previous subsection explains that faster convergence and better generalization are crucial to the test performance of over-parameterized models and these measures can be quantified by the convergence rate and sample complexity, respectively. In this subsection, we show that both quantities can be analyzed through the PL* condition and Lipschitz continuity, whose results suggest that encouraging an over-parameterized model to have a smaller condition number will generally result in faster convergence and better generalization.

**Definition 4** *(Lipschitz Continuity (Nesterov, 2003)). The function $f$ is $L_f$-Lipschitz with respect to parameters $\mathbf{w}$ if there exists a constant $L_f$, such that: $|f(\mathbf{w_2}) - f(\mathbf{w_1})| \leq L_f \|\mathbf{w_1} - \mathbf{w_2}\|, \forall \mathbf{w_1}, \mathbf{w_2} \in \mathcal{W}$, where $\mathcal{W}$ indicates the hypothesis set.*

**Definition 5** *(Polyak-Łojasiewicz\* Condition (Polyak, 1963)). The function $f(\mathbf{w})$ satisfies the Polyak-Łojasiewicz\* (PL\*) condition with parameter $\mu > 0$ if $\forall \mathbf{w} \in \mathcal{W}, \|\nabla f(\mathbf{w})\|^2 \geq \mu \cdot f(\mathbf{w})$.*

The inequality in the PL* condition implies that every stationary point is a global minimizer, which is also a property of strong convexity. However, unlike strong convexity, the PL* condition does not assume the uniqueness of the minimizer (Bassily et al., 2018). Such difference makes PL* a more appropriate mathematical framework to analyze the theoretical properties of over-parameterization as the optimization of over-parameterized models results in manifolds of global minima.

Following the original argument from Liu et al. (2022); Charles & Papailiopoulos (2018), we use the PL* condition and Lipschitz continuity to analyze the convergence rate and sample complexity of over-parameterized models trained using gradient descent (GD).

**Theorem 6** *Suppose the loss function $\mathcal{L}(\mathbf{w})$ and its first derivative are both $L_f$-Lipschitz continuous and the loss function satisfies the $\mu$-PL\* condition in the ball $B(\mathbf{w}_0, R) := \{\mathbf{w} \in \mathbb{R}^m : \|\mathbf{w} - \mathbf{w}_0\| \leq R\}$ with $R > 0$, then we have the following:*

*(a) Convergence Rate of GD (Karimi et al., 2016): Gradient descent with a step size $\eta = \frac{1}{L_f}$ converges to a global solution with an exponential convergence rate: $\mathcal{L}_S(\mathbf{w}_t) \leq (1 - \frac{\mu}{L_f})^t \mathcal{L}_S(\mathbf{w}_0)$.*

*(b) Sample Complexity: Suppose an algorithm $\mathcal{A}$ has pointwise hypothesis stability $\gamma$ with respect to a bounded loss function such that $0 \leq \mathcal{L}(\mathbf{w}) \leq M$. The model $f$ parameterized by $\mathbf{w}$ is the output of algorithm $\mathcal{A}$ trained on dataset $S$. Then, for any $\epsilon, \delta$, we have $\Pr(|R(f_S) - R_S(f_S)| \leq \epsilon) \geq 1 - \delta$, with the sample complexity $n(\epsilon, \delta) \leq \frac{6L_f^2 M}{\mu \epsilon^2 \delta} + \frac{M^2}{2\epsilon^2 \delta}$.*

The proof of the theorem is deferred to Appendix B. The theorem indicates that 1) a smaller $L_f/\mu$ would result in a faster convergence rate and decrease the convergence gap; and 2) a smaller $L_f/\mu$ would result in a smaller sample complexity, i.e., fewer samples required to obtain a generalization gap smaller than $\epsilon$ with a probability higher than $1 - \delta$. In other words, the theorem suggests that the convergence rate and generalization ability of an over-parameterized model can be improved by encouraging the model to have a smaller $L_f/\mu$, which is referred to as the condition number in (Nesterov, 2003; Gupta et al., 2021). Following the previous work, we call the ratio of the Lipschitz constant $L_f$ to the PL constant $\mu$ as the condition number.

**Definition 7** *(Condition Number (Gupta et al., 2021)). Suppose the loss function $\mathcal{L}$ satisfies the PL\* condition and has $L_f$-Lipschitz continuous gradients, the condition number is defined as $\frac{L_f}{\mu}$.*

## 3 MODEL OPTIMIZATION WITH PL CONDITION

Based on the analysis in the previous section, we advocate optimizing the condition number $\frac{L_f}{\mu}$ of over-parameterized models by adding it as a regularization term to the optimization problem; this technique is termed PL regularization. In this section, we first introduce the analytic form of PL regularization. Then, we propose a PL regularization-driven structured pruning method.

### 3.1 FORMULATION OF PL REGULARIZED OPTIMIZATION

To minimize the condition number, we propose the following regularized risk minimization problem, which adds the condition number to the training error:

$$\min_{\mathbf{w} \in \mathcal{W}} \mathcal{L}_S(\mathbf{w}) + \alpha \frac{L_f}{\mu}, \tag{4}$$

where $\mu$ indicates the PL constant of the neural network with respect to parameters $\mathbf{w}$; $L_f$ is the Lipschitz constant with respect to parameters $\mathbf{w}$; $\mathcal{L}_S(\mathbf{w})$ indicates the training error of the neural network with parameters $\mathbf{w}$; $\alpha$ is a trade-off parameter.

While it is desirable to train a network with a small condition number term, directly optimizing $\mu$ is often difficult. Following the uniform conditioning developed in previous works (Liu et al., 2022; Scaman et al., 2022) for an over-parameterized model $f$ with a Lipschitz continuous loss function, e.g., the mean square error loss, the analysis of $\mu$ could reduce to calculating and optimizing the minimal eigenvalue of the neural tangent kernel (NTK) associated to $f$. Denote the NTK matrix of the function $f$ as $\mathcal{K}(\mathbf{w}) = \mathcal{DF}(\mathbf{w})\mathcal{DF}^T(\mathbf{w}) \in \mathbb{R}^{n \times n}$, where $\mathcal{DF}$ represents the differentiable map of $f$ at $\mathbf{w}$, and the smallest eigenvalue of $\mathcal{K}$ as $\lambda_{min}(\mathcal{K}(\mathbf{w}))$; and denote $\mu^* = \inf_{\mathbf{w} \in \mathcal{W}} \lambda_{min}(\mathcal{K}(\mathbf{w}))$ based on the uniform conditioning (Liu et al., 2022; 2020a; Scaman et al., 2022). Consequently, the objective function (4) is represented as:

$$\min_{\mathbf{w} \in \mathcal{W}} \mathcal{L}_S(\mathbf{w}) + \alpha \frac{L_f}{\mu^*}. \tag{5}$$

Following the previous works (Liu et al., 2022; 2020b), the following proposition shows that the condition number is upper bounded by Hessian norm $\|H_f\|_2$, which is defined as $\|H_f\|_2 = \max_i \|H_{f_i}\|_2$, where $H_{f_i} = \frac{\partial^2 f_i}{\partial \mathbf{w}^2}$. $\|H_f\|_2$ is termed the Hessian norm of $f$ hereinafter.

**Proposition 8** *Given a point $\mathbf{w}_0$, $B(\mathbf{w}_0, R)$ denotes a ball centered at $\mathbf{w}_0$ with radius $R > 0$. Suppose the loss function satisfies the $\mu^*$-PL$^*$ condition and has $L_f$-Lipschitz gradients, for all $\mathbf{w} \in B(\mathbf{w}_0, R)$. Then the condition number is upper bounded by $\frac{L_f}{\mu^*} \leq \frac{\|\mathcal{DF}(\mathbf{w}_0)\|_2 + \sup_{\mathbf{w} \in B} \sqrt{n}R \cdot \|H_f(\mathbf{w})\|_2}{\inf_{\mathbf{w} \in B} \lambda_{min}(\mathcal{K}(\mathbf{w}))}$.*

The proof can be found in Appendix C. Proposition 8 motivates us to optimize the upper bound of condition number for PL regularization optimization.

### 3.2 IMPLEMENTATION OF PL-REGULARIZED OPTIMIZATION VIA PRUNING

In this subsection, we adopt a *pruning* approach controlled by Hessian norm to solve the new regularized risk minimization problem.

#### 3.2.1 LEARNING OBJECTIVE OF PL REGULARIZATION-DRIVEN PRUNING ALGORITHM

First, we propose the following theorem to show that pruning parameters governed by minimizing the Hessian norm $\|H_f\|_2$ is able to decrease the upper bound of the condition number.

**Theorem 9** *Suppose the neural network $f$ parameterized by $\mathbf{w}$ ($\mathbf{w} \in B(\mathbf{w}_0, R)$) and its loss function $\mathcal{L}$ have the same assumptions as in Proposition 8. Pruning parameters from the parameter set governed by minimizing the Hessian norm $\|H_f\|_2$ can decrease the upper bound of the condition number.*

**Proof sketch.** Based on the Taylor expansion around initialization $\mathbf{w}_0$, we show that the pruning parameters controlled by minimizing the Hessian norm $\|H_f\|_2$ is able to minimize the upper bound of Lipschitz (according to Proposition 8); and based on the matrix theory, pruning parameter is able to maximize the minimal eigenvalue of NTK (Proposition 14); details can be found in Appendix D.

To disable unnecessary or harmful parameters, we introduce a binary pruning mask to implement structured pruning, i.e., each element in the mask controls a subset of parameters. This design is particularly suitable for parameters with inherent structures, e.g., some parameters are from the same convolutional filter in a CNN or head in a Transformer. Suppose the parameters $\mathbf{w}$ consists of $p$ subsets of parameters and let $\mathbf{w}_{[i]}$ denote the $i^{\text{th}}$ subset of the parameters. Then the mask $\mathbf{m}$, a Boolean vector of length $p$, sparsifies the parameters as follows: $\mathbf{w}_{[i]}$ is disabled, i.e., all elements of $\mathbf{w}_{[i]}$ equal to 0, if $m_i = 0$; otherwise, $\mathbf{w}_{[i]}$ remains the same. With the introduction of the pruning mask, Eq. 5 can be transformed to the following optimization problem, which additionally learns the binary mask to prune the network:

$$\min_{\mathbf{w},\mathbf{m}} \mathcal{L}_{prune} = \mathcal{L}_S(\mathbf{m} * \mathbf{w}) + \alpha \|H_f(\mathbf{m} * \mathbf{w})\|_2, \tag{6}$$

where $*$ denotes the Khatri-Rao product, i.e., $\mathbf{m} * \mathbf{w} = \begin{bmatrix} m_1 \mathbf{w}_{[1]} & \cdots & m_p \mathbf{w}_{[p]} \end{bmatrix}^\top$; and $\|H_f(\mathbf{m} * \mathbf{w})\|_2$ denotes the Hessian norm w.r.t. the pruned model. In practice, we optimize the Hessian norm by minimizing the trace of the Hessian which can be efficiently approximated; details can be found in Appendix E.2. We further conduct an empirical analysis to show that pruning parameters controlled by penalizing the trace of Hessian leads to a decrease in the condition number; the result is included in Figure 5 of Appendix D.

Directly solving Eq. 6 to obtain $\mathbf{m}$ is difficult, and therefore we adopt a gating network to learn a binary mask and perform structured pruning. The mask is defined as a differentiable gating function: $\mathbf{m} = g(\mathbf{d_w}; \mathbf{v})$, where $\mathbf{d_w}$ and $\mathbf{v}$ are the input and parameter of the function $g$; $\mathbf{d_w}$, dubbed as the parameter features, encodes the optimization dynamics characterized by the PL$^*$ condition. In the following two subsections, we explain the details of the parameter features $\mathbf{d_w}$ and the gating function $g$. Algorithm 2 summarizes the proposed pruning algorithm.

### 3.2.2 Parameter Features

$\mathbf{d}_{\mathbf{w}_{[i]}}$ ($\mathbf{d}_{\mathbf{w}_{[i]}} \in \mathbb{R}^2$) is designed to encode two pieces of information about the optimization dynamics for each $\mathbf{w}_{[i]}$. According to the discussion in Section 3.1, the PL$^*$ condition at $\mathbf{w}$ can be analyzed through the minimum eigenvalue of the corresponding NTK. Therefore, the first parameter feature about $\mathbf{w}_{[i]}$ is chosen as $\lambda_{min}(\mathcal{K}(\mathbf{w}_{[i]}))$.

The second parameter feature is empirically set as the entropy of the eigenvalue distribution of the corresponding tangent kernel, which provides a measure to summarize the distribution envelope. Given the NTK matrix $\mathcal{K}(\mathbf{w}_{[i]})$, the entropy, denoted by $\rho$, is defined as:

$$\rho\left(\mathcal{K}\left(\mathbf{w}_{[i]}\right)\right) = -\sum_{i=1}^{n} \bar{\lambda}_i\left(\mathcal{K}\left(\mathbf{w}_{[i]}\right)\right) \log\left(\bar{\lambda}_i\left(\mathcal{K}\left(\mathbf{w}_{[i]}\right)\right)\right), \tag{7}$$

where $\bar{\lambda}_i = \lambda_i / \sum_j \lambda_j$ is the normalized eigenvalue. $\rho$ is maximal when the eigenvalues are all equal and small when a single eigenvalue is much larger than all others.

### 3.2.3 Gating Network

The gating network is implemented by a two-layer feedforward network (FN) with parameters $\mathbf{v}$, which outputs a scalar between 0 and 1 to indicate the importance of $\mathbf{w}_{[i]}$:

$$g(\mathbf{d_w}; \mathbf{v}) = \text{softmax}\left(\left[FN_{\mathbf{v}}(\mathbf{d}_{\mathbf{w}_{[1]}}); \cdots; FN_{\mathbf{v}}(\mathbf{d}_{\mathbf{w}_{[p]}})\right]\right) \in [0, 1]^p, \tag{8}$$

The parameter $\mathbf{v}$ is updated according to the pre-defined pruning times and pruning ratios $\{t_j, r_j\}_{j=1}^q$. For example, under the one-shot pruning schedule, i.e., $\{t, r\}$, we update $\mathbf{v}$ once at epoch $t$ to prune $r\%$ of parameters; under the linear pruning schedule, i.e., $\{t_j, r_f/t_{max}\}_{j=1}^{t_{max}}$ where $t_{max}$ denotes the maximum number of training iterations and $r_f$ denotes the target pruning ratio, we update $\mathbf{v}$ at every epoch to prune $r_f/t_{max}\%$ of parameters (Hoefler et al., 2021).

Once the update of $\mathbf{v}$ is complete, we apply the TopK function to generate the binary mask $\mathbf{m}$:

$$\mathbf{m} = TopK(g(\mathbf{d_w}; \mathbf{v}), k), \text{ where } m_i = \begin{cases} 1, & \text{if } g(\mathbf{d_w}; \mathbf{v})_i \text{ is in the top } k \text{ elements of } g(\mathbf{d_w}; \mathbf{v}), \\ 0, & \text{otherwise.} \end{cases}$$
$$\tag{9}$$

At pruning time $t_j$, the hyperparameter $k$ is set according to the pruning ratio: $k = (1 - r_j)p$. By utilizing such a learnable gating function, the mask is capable of adaptively inducing sparsity on the network, till the pruned network produces good accuracy and fulfills the target pruning ratio. The final output $\mathbf{m}_{t_q}$ is used to produce the sparse over-parameterized model $f(\mathbf{m}_{t_q} * \mathbf{w})$.

## 4 Related Work

**Over-parameterized Model.** Deep models used in practice are often heavily over-parameterized (Zhang et al., 2021; Nakkiran et al., 2021). Recent progress has been made in convergence analysis of over-parameterized models optimized by (stochastic) gradient descent algorithms (Du et al., 2018; Liu et al., 2020b; Oymak & Soltanolkotabi, 2019). In particular, the

PL condition (Polyak, 1963) and its slight variant – the PL* condition (Liu et al., 2022) – have attracted interest in connection with the convergence guarantees. Some works prove that for over-parameterized models with the proper random initialization, gradient descent-based methods provably find the global minimum with a global polynomial time convergence (Du et al., 2018; Belkin, 2021; Karimi et al., 2016; Liu et al., 2022). Furthermore, the work (Charles & Papailiopoulos, 2018) shows the generalization bound of models satisfying the PL condition via stability hypothesis (Bousquet & Elisseeff, 2002). Inspired by the success of recent theoretical analysis, this work uses PL* condition to analyze the convergence and generalization ability, which is then used to guide the training of practical over-parameterized models.

**Network Pruning.** Network pruning aims to minimize network parameters while maintaining the model performance. Most existing structured pruning methods rely on approximations of loss differences resulting from parameter perturbation. There are three classes of pruning techniques. The first uses zeroth-order information such as the magnitude of weight and activation values with respect to the running model, e.g., adding L1 or L2 penalty norm to the objective function (Han et al., 2015b;a; He et al., 2018) or using the Lottery Ticket Hypothesis and its extensions (Frankle & Carbin, 2018; Frankle et al., 2019; Morcos et al., 2019; You et al., 2020; Yu et al., 2020; Liu et al., 2021b). The second uses gradient-based first-order methods to approximate loss change via the first-order Taylor expansion of the training loss (Liu & Wu, 2019; You et al., 2019; Mozer & Smolensky, 1988; Molchanov et al., 2016; Sanh et al., 2020). The third uses the Hessian or Fisher information of the loss to select weights for deletion (Hassibi et al., 1993; Hassibi, 1992; Cun et al., 1990; Molchanov et al., 2019; Liu et al., 2021a). Different from structured pruning works often accompanied with negative impacts on model generalization, the proposed pruning method based on PL regularization simultaneously improves convergence and generalization ability.

## 5 EXPERIMENTS

This section presents experiments on three over-parameterized models, namely BERT (Devlin et al., 2018), Switch-Transformer (Fedus et al., 2021), and VGG-16 (Simonyan & Zisserman, 2014). Detailed ablation experiments can be found in Appendix F.

**Evaluation Measure.** We evaluate the effectiveness of our algorithm for model optimization in terms of *training efficiency* and *generalization ability*. Training efficiency is measured by the training error of the model trained with fixed training iterations (or epochs) (Shazeer et al., 2017; Fedus et al., 2021). Generalization ability is measured by the test performance (Liu et al., 2021b). For fair comparisons, all models are evaluated with the same pruning ratio, which is computed as the number of disabled parameters divided by the parameter counts of the corresponding structure.

### 5.1 RESULTS ON BERT

This experiment focuses on the Multi-Head Attention (MHA) component of BERT-Base (Devlin et al., 2018), as MHA plays a critical role in the success of BERT. Following the basic setup of (Devlin et al., 2018), we train BERT with self-supervised masked language modeling (MLM) task (Devlin et al., 2018) on WikiText-2 (Merity et al., 2016). The baseline methods contain vanilla BERT (Wang et al., 2019), two Transformer-based pruning methods, namely BERT-LTH (Chen et al., 2020) and Att-Score (Michel et al., 2019), and two pruning approaches, namely SNIP (Lee et al., 2018) and GraSP (Wang et al., 2020). More experimental details can be found in Appendix E.3.

Table 1: Results of BERT on WikiText-2. We denote perplexity as PPL, and $\Delta$PPL represents the perplexity difference between training and test. Boldface indicates the best result.

| Method | Criterion | Training PPL | | | Test PPL | $\Delta$PPL |
|--------|-----------|------------|------------|------------|----------|-------|
| | | @8 k iters | @10 k iters | @15 k iters | Final | |
| BERT | - | 39.37 | 14.79 | 6.35 | 75.57 | 7.49 |
| BERT-LTH | Magnitude | 36.97 | 13.07 | 7.92 | 81.21 | 7.20 |
| Att-Score | Gradient | 23.10 | 11.35 | 5.53 | 79.84 | 7.36 |
| SNIP | Gradient | 16.33 | 9.91 | 4.32 | 72.39 | 8.09 |
| GraSP | Hessian | 39.96 | 12.22 | 6.28 | 81.61 | 8.21 |
| Ours | PL | **15.72** | **9.63** | **4.24** | **63.18** | **6.84** |

Table 1 shows the performance of all models on WikiText-2 with the $\{5, 75\%\}$ pruning policy, i.e., we prune 75% heads at epoch-5 during training. Our method shows better training efficiency and generalization ability than all baselines. In particular, our method improves test perplexity over vanilla BERT, and improves training perplexity by 60%, 29%, and 33% after 8 k, 10 k, and 15 k iterations, demonstrating the effectiveness of PL regularization for BERT model optimization. We also evaluate the training efficiency by measuring the number of iterations required for converging to a fixed training error status. Compared to vanilla BERT, our method requires only $0.7\times$ iterations to achieve satisfactory test perplexity (see Appendix F.2). Other pruning methods, on the other hand, can improve training efficiency but at a cost of sacrificing test performance, as their pruning criteria focus on training loss, ignoring the measurement of generalization ability.

Furthermore, we analyze the effect of pruning times of mask for BERT optimization (shown in Appendix F.3). It is interesting to find that compared with the model applying PL regularization iteratively throughout training, the model that applies PL regularization during the early stages of training has better training efficiency and test performance. This phenomenon indicates that important heads for BERT optimization can potentially be discovered in the early stages of training.

## 5.2 RESULTS ON SWITCH-TRANSFORMER

Switch-Transformer (Fedus et al., 2021) is a representative implementation of Mixture-of-Expert (MoE) architecture (Shazeer et al., 2017), which has shown excellent performance in NLP tasks recently. For comparison, the baseline methods include vanilla Switch-Transformer, Switch-LTH (Chen et al., 2020), Exp-Score (Michel et al., 2019), SNIP (Lee et al., 2018), and GraSP (Wang et al., 2020). All experiments are conducted on WikiText-103 (Merity et al., 2016) with the masked language modeling task. More experimental details can be found in Appendix E.3.

Table 2: Results of Switch-Transformer on WikiText-103 (Best results in Boldface).

| Method | Training PPL | | | Test PPL | $\Delta$PPL |
| --- | --- | --- | --- | --- | --- |
| | @50 k iters | @60 k iters | @70 k iters | Final | |
| Switch-Transformer | 18.62 | 7.50 | 7.27 | 8.96 | 1.56 |
| Switch-LTH | 19.97 | 10.08 | 7.61 | 10.72 | 1.59 |
| Exp-Score | 18.45 | 10.56 | 8.17 | 9.18 | 1.49 |
| SNIP | 12.12 | 8.99 | 7.99 | 9.88 | 1.46 |
| GraSP | 18.93 | 8.17 | 7.81 | 8.84 | 1.52 |
| Ours (experts only) | 7.11 | **5.62** | 5.53 | **7.68** | 1.51 |
| Ours (experts & heads) | **6.97** | 5.86 | **5.23** | 7.78 | **1.45** |

Table 2 lists the performance of Switch-Transformer on WikiText-103 with the $\{5, 50\%\}$ pruning policy. Our method (Ours (experts only)) significantly outperforms all the baselines in both training efficiency and generalization ability, demonstrating our method can be applicable to MoE architecture. In particular, compared with the vanilla Switch-Transformer, when used on the MoE architecture, our method improves training perplexity by 62%, 41%, 24% in 50 k, 60 k, and 70 k iterations, while improving test performance by 13%. This phenomenon indicates that while the sparsely-activated MoE architecture selects different experts for each data point, there also exist certain poorly-behaved experts. By modeling the optimization dynamics which guarantee the convergence and generalization ability, the proposed PL regularization-driven pruning method can disable under- or over-specialized experts and keep well-behaved experts.

We also investigate the benefits of implementing PL regularization on both heads of MHA and experts of MoE, as shown in the last line in Table 2. Compared with the vanilla Switch-Transformer, implementing PL regularization on both heads and experts presents a 28% improvement in training perplexity after 70 k iterations and a 13% improvement in test performance. This result further demonstrates the proposed PL regularization is flexible and applicable to different architectures.

## 5.3 RESULTS ON VGG-16

This experiment focuses on VGG-16 (Simonyan & Zisserman, 2014) trained on CIFAR-10 and CIFAR-100 datasets. The baselines include vanilla VGG-16 (Li et al., 2016) three filter pruning methods, namely FPGM (He et al., 2019), CHIP (Sui et al., 2021), and DPFPS (Ruan et al., 2021), and three general pruning approaches, namely LTH (Frankle & Carbin, 2018), SNIP (Lee et al.,

2018), and GraSP (Wang et al., 2020). All models are trained from scratch and are pruned with a linear schedule where starting from epoch-15, we prune the same number of filters at each epoch until the target sparsity is reached. More details of baselines and experimental settings can be found in Appendix E.4. Additional experimental results of ResNet-56 can be found in Appendix F.1.

Table 3 shows the performance of all methods trained on CIFAR-10 and CIFAR-100 with a $50\%$ pruning ratio. As we can see, our method outperforms all pruning baselines in terms of training efficiency and test accuracy on both datasets. Especially, compared with the vanilla VGG-16, our method achieves a higher accuracy with fewer parameters, demonstrating the effectiveness of PL regularization implemented as structured pruning for VGG-16 optimization.

Table 3: Results of VGG-16 on CIFAR-10 and CIFAR-100 (Best results in Boldface).

| Method & Dataset | CIFAR-10 | | | | | CIFAR-100 | | | | |
|---|---|---|---|---|---|---|---|---|---|---|
| | Train Loss (@ epochs) | | Test Accuracy (@ epochs) | | | Train Loss (@ epochs) | | Test Accuracy (@ epochs) | | |
| | @100 | @150 | @100 | @150 | Final | @100 | @150 | @100 | @150 | Final |
| VGG-16 | 0.1091 | 0.0075 | 92.46 | 93.25 | 93.85 | 0.4826 | 0.0459 | 72.21 | 72.28 | 73.86 |
| LHT | 0.1206 | 0.0106 | 92.41 | 93.12 | 93.59 | 0.5342 | 0.0507 | 71.78 | 71.86 | 73.11 |
| SNIP | 0.1106 | 0.0098 | 92.35 | 93.21 | 93.85 | - | - | - | - | - |
| FPGM | 0.1144 | 0.0146 | 90.80 | 91.70 | 93.50 | 0.4964 | 0.0709 | 68.00 | 68.30 | 68.90 |
| GraSP | 0.2150 | 0.0420 | 91.13 | 92.83 | 93.70 | 0.8380 | 0.1240 | 69.33 | 71.74 | 72.92 |
| CHIP | 0.2585 | 0.0841 | 87.00 | 90.38 | 93.71 | 1.0581 | 0.4508 | 58.44 | 67.28 | 72.80 |
| DPFPS | 0.1224 | 0.0101 | 92.54 | 93.11 | 93.99 | 0.5222 | 0.0430 | 71.69 | 73.17 | 73.88 |
| Ours | **0.1040** | **0.0068** | **92.70** | **93.55** | **94.27** | 0.5175 | **0.0428** | 72.31 | 73.26 | 74.08 |

## 5.4 QUANTITATIVE ANALYSIS

This experiment aims to understand why the baselines perform differently in terms of training efficiency and test performance. Theorem 6 states that the condition number controls the convergence and generalization ability of a model. Thus, we take a trained Switch-Transformer as an example and compare the condition number of important experts identified by different pruning criteria, including the weight magnitude of LTH, the gradient-based score of Exp-Score, the gradient value of SNIP, and the Hessian value of GraSP.

Figure 2 presents heatmaps of the condition number and marks important experts by squares. It shows that our method has the ability to retain well behaved experts with small

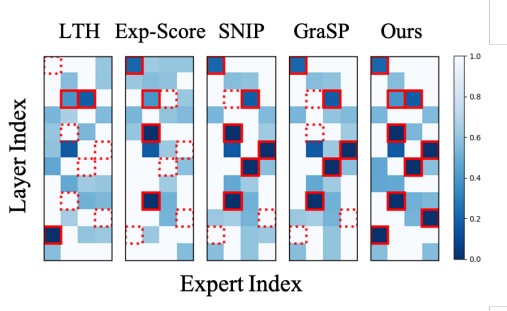

Figure 2: Heatmaps of condition number of the remained experts determined by LTH, Exp-Score, SNIP, GraSP, and our method. Baselines fail to identify the well-behaved experts (The experts marked in red are the ones that are well-behaved but missed by baselines).

condition number while the baselines fail to retain these experts. The reason is that these baselines bias towards training loss, adversely affecting generalization. This observation confirms the importance of simultaneously considering both the convergence gap and the generalization gap for over-parameterized model optimization. Furthermore, the criteria of recent pruning works can be regarded as the parameter features and fed into the proposed gating network, which may help optimize PL regularization by providing rich information of parameters. We leave it for future works.

## 6 CONCLUSION

This paper establishes a principled connection between model optimization and the PL condition and advocates utilizing this theoretical finding when optimizing over-parameterized models. A new optimization problem is therefore formulated with the proposed PL regularization, and we illustrate that it can be solved via a structured pruning method. We postulate that the effect of such regularization may be achieved in other ways, e.g., by using more advanced optimization algorithms which induce implicit regularization. Moreover, experiments demonstrate the superiority of the proposed PL regularization-driven pruning algorithm, unveiling the potential of adopting theoretically founded measures as the pruning criteria.

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

## A    BAYES ERROR DECOMPOSITION

Motivated by the previous work (Gühring et al., 2020), the Bayes error, i.e., the optimal test error, can be decomposed as:

$$R(f^*) \leq \underbrace{R_S(\hat{f}_{\mathcal{H},S})}_{\text{empirical error}} + \underbrace{|R_S(f^*_{\mathcal{H},S}) - R_S(\hat{f}_{\mathcal{H},S})|}_{\text{convergence gap}} + \underbrace{|R(f^*_{\mathcal{H},S}) - R_S(f^*_{\mathcal{H},S})|}_{\text{generalization gap}}$$
$$+ \underbrace{|R(f^*_{\mathcal{H}}) - R(f^*_{\mathcal{H},S})|}_{\text{estimation error}} + \underbrace{|R(f^*_{\mathcal{H}}) - R(f^*)|}_{\text{approximation error}}. \tag{10}$$

In the over-parameterized setting, we assume that the optimal function $f^*_{\mathcal{H},S}$ fits or interpolates the training data exactly, i.e., $R_S(f^*_{\mathcal{H},S}) = 0$. Thus the empirical error equals the convergence gap. Moreover, the estimation error can be associated with the generalization gap: $|R(f^*_{\mathcal{H},S}) - R_S(f^*_{\mathcal{H},S})| \leq 2 \sup_{f \in \mathcal{H}} |R(f) - R_S(f)|$ (Bousquet et al., 2003). Furthermore, as over-parameterization refers to manifolds of potential interpolating predictors, $\mathcal{H}$ for the over-parameterized model can be omitted, i.e., $R(f^*_{\mathcal{H}}) = R(f^*)$. Therefore, for over-parameterized models, Eq. 10 can be simplified as

$$R(f^*) \leq 2 \underbrace{|R_S(\hat{f}_{\mathcal{H},S}) - R_S(f^*_{\mathcal{H},S})|}_{0} + 3 \sup_{f \in \mathcal{H}} \underbrace{|R(f) - R_S(f)|}_{\text{generalization gap}}. \tag{11}$$
$$\underbrace{\phantom{R(f^*) \leq 2 |R_S(\hat{f}_{\mathcal{H},S}) - R_S(f^*_{\mathcal{H},S})|}}_{\text{convergence gap}}$$

## B    PROOF OF THEOREM 6

### B.1    CONVERGENCE RATE

The convergence rate of the over-parameterized model has been studied by many works (Liu et al., 2022; Karimi et al., 2016; Allen-Zhu et al., 2019).

We assume that the loss function has the $L_f$-Lipschitz continuous gradient, i.e., for all parameters $\mathbf{w}$ and $\mathbf{v}$, such that:

$$\mathcal{L}(\mathbf{w}) \leq \mathcal{L}(\mathbf{v}) + \langle \nabla \mathcal{L}(\mathbf{v}), \mathbf{w} - \mathbf{v} \rangle + \frac{L_f}{2} \|\mathbf{w} - \mathbf{v}\|^2, \tag{12}$$

Using the assumption, we have that:

$$\mathcal{L}_S(\mathbf{w}_{t+1}) - \mathcal{L}_S(\mathbf{w}_t) \leq \langle \nabla \mathcal{L}_S(\mathbf{w}_t), \mathbf{w}_{t+1} - \mathbf{w}_t \rangle + \frac{L_f}{2} \|\mathbf{w}_{t+1} - \mathbf{w}_t\|^2 \tag{13}$$

Given the gradient method with a learning rate $\eta = \frac{1}{L_f}$, such that $\mathbf{w}_{t+1} = \mathbf{w}_t - \eta \nabla \mathcal{L}_S(\mathbf{w}_t) = \mathbf{w}_t - \frac{1}{L_f} \nabla \mathcal{L}_S(\mathbf{w}_t)$, we have

$$\mathcal{L}_S(\mathbf{w}_{t+1}) - \mathcal{L}_S(\mathbf{w}_t) \leq -\frac{1}{2L_f} \|\nabla \mathcal{L}_S(\mathbf{w}_t)\|^2 \tag{14}$$

Under the PL* condition at point $\mathbf{w}_t$, we have

$$\mathcal{L}_S(\mathbf{w}_{t+1}) - \mathcal{L}_S(\mathbf{w}_t) \leq -\frac{\mu}{L_f}(\mathcal{L}_S(\mathbf{w}_t)) \tag{15}$$

Therefore, we have

$$\mathcal{L}_S(\mathbf{w}_t) \leq (1 - \frac{\mu}{L_f})(\mathcal{L}_S(\mathbf{w}_{t-1})) \leq (1 - \frac{\mu}{L_f})^t(\mathcal{L}_S(\mathbf{w}_0)) \tag{16}$$

### B.2    SAMPLE COMPLEXITY

Following previous work (Charles & Papailiopoulos, 2018), we first calculate the generalization bound $\epsilon$ for the function $f$ with the pointwise hypothesis stability $\gamma$. Then, we analyze the pointwise hypothesis stability $\gamma$ for model trained on the loss function satisfying the PL* condition with parameter $\mu$. Finally, given the pointwise hypothesis stability $\gamma$, and the generalization bound $\epsilon$, we bound the sample complexity of $f$.

### B.2.1 GENERALIZATION ANALYSIS VIA POINTWISE HYPOTHESIS STABILITY

A useful approach to analyzing the generalization performance of learning algorithms is algorithmic stability (Bousquet & Elisseeff, 2002). A learning algorithm $\mathcal{A}$ is stable if small changes in the training set result in small differences in the output predictions of the trained model. Following the foundational works, we analyze the generalization bound from the perspective of *pointwise hypothesis stability*.

Given a data set $S = \{\mathbf{x}_1, \ldots, \mathbf{x}_n\}$ where $\mathbf{x}_i \sim \mathcal{D}$, we define dataset $S^i$ as $S \backslash \mathbf{x}_i$, i.e., $S^i = \{\mathbf{x}_1, \ldots, \mathbf{x}_{i-1}, \mathbf{x}_{i+1}, \ldots, \mathbf{x}_n\}$. For our purposes, $f$ is the output of learning algorithm $\mathcal{A}$, parameterized by $\mathbf{w}$. For the model $f$ parameterized by $\mathbf{w}$, we denote the model trained on the dataset $S$ and $S^i$ as $\mathbf{w}_S$ and $\mathbf{w}_{S^i}$, respectively. Accordingly, the loss functions are represented as $\mathcal{L}(\mathbf{w}_S; \mathbf{x})$ and $\mathcal{L}(\mathbf{w}_{S^i}; \mathbf{x})$, respectively. Especially, for a given dataset $S$, the empirical loss on a dataset $S$ is denoted as $\mathcal{L}_S(\mathbf{w}_S)$ and given by $\mathcal{L}_S(\mathbf{w}_S) = \frac{1}{n} \sum_{i=1}^{n} \mathcal{L}(f(\mathbf{x}_i; \mathbf{w}_S), \mathbf{y}_i))$.

**Definition 10** *(Pointwise Hypothesis Stability (Bousquet & Elisseeff, 2002)). An algorithm $\mathcal{A}$ has pointwise hypothesis stability $\gamma$ with respect to a loss function $\mathcal{L}$ if*

$$\forall i \in \{1, \ldots, n\}, \quad \mathbb{E}_S\left[|\mathcal{L}(\mathbf{w}_S; \mathbf{x}_i) - \mathcal{L}(\mathbf{w}_{S^i}; \mathbf{x}_i)|\right] \leq \gamma. \tag{17}$$

Then we can use the pointwise hypothesis stability to establish the generalization bounds as

**Theorem 11** *Bousquet & Elisseeff (2002). Suppose we have a learning algorithm $\mathcal{A}$ with pointwise hypothesis stability $\gamma$ with respect to a bounded loss function $\mathcal{L}$ such that $0 \leq \mathcal{L}(\mathbf{w}; \mathbf{x}) \leq M$. For any $\delta$, we have with probability at least $1 - \delta$,*

$$R(\mathbf{w}_S) \leq R_S(\mathbf{w}_S) + \sqrt{\frac{M^2 + 12Mn\gamma}{2n\delta}}. \tag{18}$$

In the following, we derive stability bounds for models trained on risk functions satisfying the PL$^*$ condition.

### B.2.2 POINTWISE HYPOTHESIS STABILITY WITH PL$^*$ CONDITION

**Theorem 12** *Assume that for all training sets $S$ and models $f$ parameterized by $\mathbf{w}$, the loss function $\mathcal{L}$ is PL$^*$ with parameter $\mu$. In addition, assume that the over-parameterized model $f$ with parameter $\mathbf{w}_S$ trained on $S$ is capable of converging to some global minimizer $\mathbf{w}_S^*$. Then for all $S$, if $|\mathcal{L}_S(\mathbf{w}_S) - \mathcal{L}_S(\mathbf{w}_S^*)| \leq \epsilon_f$, then $f$ has pointwise hypothesis stability with parameter $\gamma$ as:*

$$\gamma \leq \frac{L_f^2}{\mu n}. \tag{19}$$

*Proof.* For a training set $S$, let $\mathbf{w}_S$ denote the parameters of $f$ on $S$, and let $\mathbf{w}_{S^i}$ denote the parameters of $f$ on $S^i$, where $S^i$ denotes $S \backslash \mathbf{x}_i$. Let $\mathbf{w}_S^*$ and $\mathbf{w}_{S^i}^*$ denote the corresponding optimal solutions, respectively. We then have,

$$|\mathcal{L}(\mathbf{w}_S; \mathbf{x}_i) - \mathcal{L}(\mathbf{w}_{S^i}; \mathbf{x}_i)|$$
$$\leq |\mathcal{L}(\mathbf{w}_S; \mathbf{x}_i) - \mathcal{L}(\mathbf{w}_S^*; \mathbf{x}_i)| + |\mathcal{L}(\mathbf{w}_S^*; \mathbf{x}_i) - \mathcal{L}(\mathbf{w}_{S^i}^*; \mathbf{x}_i)| + |\mathcal{L}(\mathbf{w}_{S^i}^*; \mathbf{x}_i) - \mathcal{L}(\mathbf{w}_{S^i}; \mathbf{x}_i)|. \tag{20}$$

The three terms can be separately bounded.

**The first and third term.**

As discussed in previous work (Karimi et al., 2016), the PL$^*$ condition implies that the quadratic growth (QG) condition

$$\frac{\mu}{2}\|\mathbf{w}_S - \mathbf{w}_S^*\|^2 \leq |\mathcal{L}_S(\mathbf{w}_S) - \mathcal{L}_S(\mathbf{w}_S^*)|. \tag{21}$$

Using the fact that $|\mathcal{L}_S(\mathbf{w}_S) - \mathcal{L}(\mathbf{w}_S^*)| \leq \epsilon_f$ by the assumption, it implies

$$\|\mathbf{w}_S - \mathbf{w}_S^*\| \leq \frac{\sqrt{2}}{\sqrt{\mu}}\sqrt{|\mathcal{L}_S(\mathbf{w}_S) - \mathcal{L}_S(\mathbf{w}_S^*)|} = \sqrt{\frac{2\epsilon_f}{\mu}}. \tag{22}$$

Since that loss function $\mathcal{L}$ is $L_f$-Lipschitz, we can bound the first and third term as:

$$|\mathcal{L}(\mathbf{w}_S; \mathbf{x}_i) - \mathcal{L}(\mathbf{w}_S^*; \mathbf{x}_i)| \leq L_f \|\mathbf{w}_S - \mathbf{w}_S^*\| \leq L_f \sqrt{\frac{\epsilon_f}{\mu}};$$

$$|\mathcal{L}(\mathbf{w}_{S^i}; \mathbf{x}_i) - \mathcal{L}(\mathbf{w}_{S^i}^*; \mathbf{x}_i)| \leq L_f \|\mathbf{w}_{S^i} - \mathbf{w}_{S^i}^*\| \leq L_f \sqrt{\frac{\epsilon_f}{\mu}};$$

(23)

In the over-parameterized settings, the model $f$ optimized by the loss function enables a global optimizer $\mathbf{w}^*$ and has a small $\epsilon_f$ constant, thus the first term and third term equal to 0.

**The second term.** PL$^*$ condition implies that $\mathcal{L}(\mathbf{w}_S^*; \mathbf{x}_i) = 0$, and $\mathcal{L}_{S^i}(\mathbf{w}_{S^i}^*) = 0$ thus the second term can be manipulated as:

$$\begin{aligned}
|\mathcal{L}(\mathbf{w}_S^*; \mathbf{x}_i) - \mathcal{L}(\mathbf{w}_{S^i}^*; \mathbf{x}_i)| &= |\mathcal{L}(\mathbf{w}_{S^i}^*; \mathbf{x}_i)| \\
&= |n\mathcal{L}_S(\mathbf{w}_{S^i}^*) - (n-1)\mathcal{L}_{S^i}(\mathbf{w}_{S^i}^*)| \\
&= n|\mathcal{L}_S(\mathbf{w}_{S^i}^*)|.
\end{aligned}$$

(24)

Note that since $\nabla\mathcal{L}_{S^i}(w_{S^i}^*) = 0$, we get:

$$\|\nabla\mathcal{L}_S(\mathbf{w}_{S^i}^*)\|^2 = \frac{1}{n^2}\|\nabla\mathcal{L}(\mathbf{w}_{S^i}^*; \mathbf{x}_i)\|^2 \leq \frac{L_f^2}{n^2}.$$

(25)

Furthermore, PL$^*$ condition implies $\mu\mathcal{L}_S(\mathbf{w}_{S^i}^*) \leq \|\nabla\mathcal{L}_S(\mathbf{w}_{S^i}^*)\|^2$, we can obtain

$$n\,|\mathcal{L}_S(\mathbf{w}_{S^i}^*)| \leq \frac{n}{\mu}\|\nabla\mathcal{L}_S(\mathbf{w}_{S^i}^*)\|^2 \leq \frac{n}{\mu}\frac{L_f^2}{n^2} \leq \frac{L_f^2}{\mu n}.$$

(26)

**The overall bound.** Plugging Eqs. 23 and 26 into Eq. 20, we can obtain the desired result: Assume that for all $S$ and $\mathbf{w}$, the loss function $\mathcal{L}$ is PL$^*$ with parameter $\mu$, algorithm $\mathcal{A}$ has pointwise hypothesis stability with parameter $\gamma$ as:

$$\gamma \leq \frac{L_f^2}{\mu n}.$$

(27)

∎

### B.2.3 BOUND OF SAMPLE COMPLEXITY

Based on the above subsection, if the loss function $\mathcal{L}$ of model $f$ is PL$^*$ condition with parameter $\mu$, and the algorithm $\mathcal{A}$ has the pointwise hypothesis stability with parameter $\gamma \leq \frac{L_f^2}{\mu n}$, the generalization gap $\epsilon$ of the obtained model $f$ can be bound as: $\epsilon = \sqrt{\frac{M^2 + 12Mn\gamma}{2n\delta}}$. Therefore, we can compute the sample complexity as:

$$n(\epsilon, \delta) \leq \frac{6L_f^2 M}{\mu\epsilon^2\delta} + \frac{M^2}{2\epsilon^2\delta}$$

(28)

## C   PROOF OF PROPOSITION 8

*Proof.* Consider an arbitrary point $\mathbf{w} \in B(\mathbf{w}_0, R)$. For all sample $i$ in dataset $S$, we have

$$\mathcal{DF}_i(\mathbf{w}) = \mathcal{DF}_i(\mathbf{w}_0) + \int_0^1 H_{f_i}(\mathbf{w}_0 + \tau(\mathbf{w} - \mathbf{w}_0))(\mathbf{w} - \mathbf{w}_0)\,d\tau.$$

(29)

Since $\tau \in [0, 1]$, the point $\mathbf{w}_0 + \tau(\mathbf{w} - \mathbf{w}_0)$ is inside of the ball, thus $\|H_{f_i}(\mathbf{w}_0 + \tau(\mathbf{w} - \mathbf{w}_0))\|_2 \leq \|H_f(\mathbf{w})\|_2$. Therefore, Eq. 29 can be bounded as:

$$\begin{aligned}
\|\mathcal{DF}_i(\mathbf{w}) - \mathcal{DF}_i(\mathbf{w}_0)\|_2 &\leq \sup_{\tau \in [0,1]}\|H_{f_i}(\mathbf{w}_0 + \tau(\mathbf{w} - \mathbf{w}_0))\|_2 \cdot \|\mathbf{w} - \mathbf{w}_0\|_2 \\
&\leq R \cdot \|H_f(\mathbf{w})\|_2.
\end{aligned}$$

(30)

By triangle inequality, we have,

$$
\begin{aligned}
\|\mathcal{D}\mathcal{F}(\mathbf{w})\|_2 - \|\mathcal{D}\mathcal{F}(\mathbf{w}_0)\|_2 &\leq \|\mathcal{D}\mathcal{F}(\mathbf{w}) - \mathcal{D}\mathcal{F}(\mathbf{w}_0)\|_2 \\
&\leq \|\mathcal{D}\mathcal{F}(\mathbf{w}) - \mathcal{D}\mathcal{F}(\mathbf{w}_0)\|_F \\
&= \sqrt{\sum_i \|\mathcal{D}\mathcal{F}_i(\mathbf{w}) - \mathcal{D}\mathcal{F}_i(\mathbf{w}_0)\|^2} \\
&\leq \sqrt{n}R \cdot \|H_f(\mathbf{w})\|_2.
\end{aligned}
\tag{31}
$$

By re-arranging Eq. 31, we have

$$
\|\mathcal{D}\mathcal{F}(\mathbf{w})\|_2 \leq \|\mathcal{D}\mathcal{F}(\mathbf{w}_0)\|_2 + \sqrt{n}R \cdot \|H_f(\mathbf{w})\|_2.
\tag{32}
$$

Using the definition of $L_f$, we have

$$
L_f = \sup_{\mathbf{w}\in B} \|D\mathcal{F}(\mathbf{w})\|_2 \leq \|\mathcal{D}\mathcal{F}(\mathbf{w}_0)\|_2 + \sqrt{n}R \cdot \sup_{\mathbf{w}\in B} \|H_f(\mathbf{w})\|_2.
\tag{33}
$$

Combining Eq. 33 and definition of $\mu^*$, we have

$$
\frac{L_f}{\mu^*} \leq \frac{\|\mathcal{D}\mathcal{F}(\mathbf{w}_0)\| + \sup_{\mathbf{w}\in B} \|H_f(\mathbf{w})\| \cdot R}{\inf_{\mathbf{w}\in B} \lambda_{min}(\mathcal{K}(\mathbf{w}))}.
\tag{34}
$$

$\blacksquare$

## D  PROOF OF THEOREM 9

To prove that minimizing the upper bound of condition number can be achieved by pruning parameters controlled by Hessian norm $\|H_f\|_2$, we separately analyze the impact of pruning parameters controlled by Hessian norm $\|H_f\|_2$ on the Lipschitz constant; and the impact of pruning parameters on the minimal eigenvalue of NTK (Proposition 14).

*Proof.* Using the same argument in Proposition 8, as shown in Eq. 33 where

$$
L_f = \sup_{\mathbf{w}\in B} \|D\mathcal{F}(\mathbf{w})\|_2 \leq \|\mathcal{D}\mathcal{F}(\mathbf{w}_0)\|_2 + \sqrt{n}R \cdot \sup_{\mathbf{w}\in B} \|H_f(\mathbf{w})\|_2,
$$

we can conclude that pruning parameters reduces the parameter size, leading to the decrease of the $\|\mathbf{w} - \mathbf{w}_0\|$ (i.e., the radius of the ball, $R$); and pruning parameters governed by minimizing the Hessian norm $\|H_f\|_2$ can decrease the upper bound of the Lipschitz constant. In summary, the pruning operation with controlling of Hessian norm is able to minimize the upper bound of $L_f$. $\blacksquare$

Then we use matrix analysis to show that maximizing the minimal eigenvalue of NTK can be achieved via pruning.

Let us denote the matrix $\mathbf{G} = \mathcal{D}\mathcal{F}\mathcal{D}\mathcal{F}^\top$. The matrix $\mathbf{G}$ and NTK matrix ($\mathcal{K} = \mathcal{D}\mathcal{F}^\top\mathcal{D}\mathcal{F}$) shares the same non-zero eigenvalues, thus we analyze the effect of pruning on the eigenvalues of NTK via the matrix $\mathbf{G} \in \mathbb{R}^{m\times m}$ where $m$ denotes the parameter size.

Pruning any parameter $w_p$ from the weight set $\mathbf{w}$ corresponds to deleting the corresponding $p$-th row and the corresponding $p$-th column from $\mathbf{G}$, i.e, $\tilde{\mathbf{G}}_p$. According to the Cauchy Interlacing Theorem, increasing the minimal eigenvalue of $\mathbf{G}$ can be achieved by pruning the parameter $w_m$. We first summarize the Cauchy Interlacing Theorem as follows:

**Theorem 13** *(Cauchy Interlacing Theorem Horn & Johnson (2012)). Let $B \in M_m$ be Hermitian, let $y \in \mathbf{C}^m$ and $a \in \mathbf{R}$ be given, and let $A = \begin{bmatrix} B & y \\ y^* & a \end{bmatrix} \in M_{m+1}$. Then*

$$
\lambda_1(A) \leq \lambda_1(B) \leq \lambda_2(A) \leq \cdots \leq \lambda_m(A) \leq \lambda_m(B) \leq \lambda_{m+1}(A).
\tag{35}
$$

Theorem 13 allows us to conclude that if we remove the parameter $w_m$, the corresponding $\tilde{\mathbf{G}}_m \subseteq \mathbf{G}$, then $\lambda_{min}(\tilde{\mathbf{G}}_m) \geq \lambda_{min}(\mathbf{G})$.

**Proposition 14** *For an over-parameterized model $f$ parameterized by $\mathbf{w} \in \mathcal{W}$, pruning operation on the parameter set $\mathbf{w}$ increases the minimal eigenvalue of NTK matrix associated to $f$.*

*Proof.* For parameter $w_p$ and matrix $\mathbf{G}$, let $\mathbf{G}^{p \to m}$ denote the permuted matrix obtained by permuting $p$-th row and $p$-th column of matrix $\mathbf{G}$ to $m$-th row and $m$-th column. Then pruning parameter $w_p$ is equivalent to pruning $w_m$ of $\mathbf{G}^{p \to m}$ matrix.

Following Theorem 13, pruning parameter $w_m$ of $\mathbf{G}^{p \to m}$ will increase its minimal eigenvalue. Moreover, using the fact that $\mathbf{G}^{p \to m} = \mathbf{P}\mathbf{G}\mathbf{P}^\top = \mathbf{P}\mathbf{G}\mathbf{P}^{-1}$ where $\mathbf{P}$ denotes the permutation matrix, we have $\mathbf{G}^{p \to m}$ and $\mathbf{G}$ are similar matrix with same eigenvalues. Thus, the minimal eigenvalue of $\mathbf{G}$ is increased. In summary, pruning any parameter $w_p$ will increase the minimal eigenvalue of $\mathbf{G}$. ∎

*Proof of Theorem 9.* Proposition 8 shows that pruning operation controlled by minimizing Hessian norm $\|H_f\|_2$ is able to minimize the upper bound of Lipschitz constant $L_f$; and Proposition 14 shows that pruning parameter is able to maximize the minimal eigenvalue of NTK. Combining these two propositions, we can conclude that pruning parameters controlled by minimizing the Hessian norm $\|H_f\|_2$ is able to minimize the condition number of the model. ∎

To further verify the claim of theories, we empirically investigate the evolution of condition number of the pruned model and vanilla model. We apply a small BERT with 4 Transformer layers to WikiText-2 for the language modeling task. We prune the heads of vanilla BERT by using a strategy of pruning 5% heads every 10 epochs, controlled by minimizing the Hessian trace (details can be found in Appendix E.2); the pruned network is denoted as Masked-BERT. As shown in Figure 3, compared with vanilla BERT, Masked-BERT presents a smaller value of condition number [1]. Especially, we can observe a sharp decrease in the condition number of Masked-BERT after each pruning operation; to highlight this, red lines are included in the figure which indicate the condition number before and after pruning.

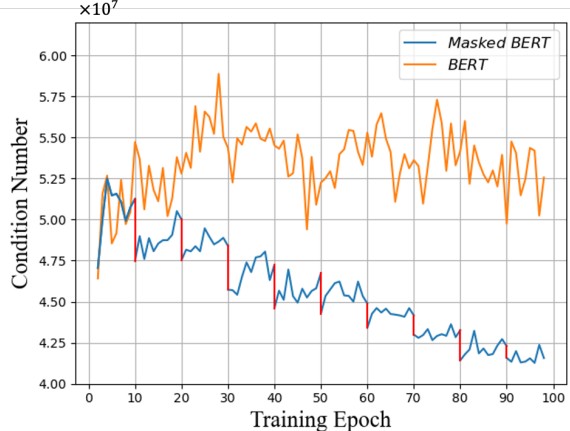

Figure 3: The evolution of condition number of the vanilla BERT and the pruned BERT (denoted as Masked-BERT).

# E EXPERIMENTAL DETAILS

## E.1 ALGORITHM OVERVIEW

Figure 4 shows the overview of the proposed method. We impose PL regularization for model optimization by adopting a pruning approach. More specifically, we introduce a binary mask for periodically sparsifying parameters, and the mask is learned via a gating network whose input summarizes

---

[1] Condition number is computed as $\frac{\lambda_{max}(\mathcal{K})}{\lambda_{min}(\mathcal{K})}$, where $\lambda_{max}$ and $\lambda_{min}$ denote the largest and smallest eigenvalue, respectively.

the optimization dynamics of sub-networks in terms of the PL condition. Algorithm 2 summarizes the proposed pruning algorithm.

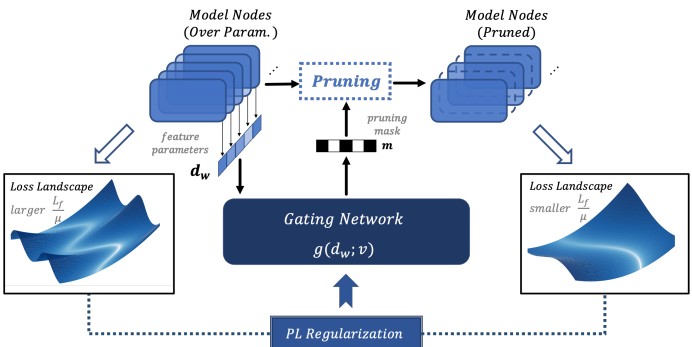

Figure 4: Algorithm overview. The gating network generates binary mask $\mathbf{m}$ based on the parameter features $\mathbf{d_w}$. PL Regularization helps pruned model obtain a smaller condition number $\frac{L_f}{\mu}$.

### E.2 IMPLEMENTATION OF LEARNING OBJECTIVE

Based on the Theorem 9, minimizing the upper bound of condition number can be achieved by pruning parameters controlled by minimizing the Hessian norm.

In practice, we minimize the Hessian norm by minimizing the trace of the corresponding Hessian where $Trace(H_f(\mathbf{w})) = \sum_i^n Trace(H_i(\mathbf{w}))$. Consequently, the learning objective of the gating network is approximated as:

$$\min_{\mathbf{w},\mathbf{v}} \mathcal{L}_S(\mathbf{m} * \mathbf{w}) + \alpha Trace\left(H_f\left(g\left(\mathbf{d_w}; \mathbf{v}\right) * \mathbf{w}\right)\right). \tag{36}$$

Specifically, we use the Hutchinson method (Sankar et al., 2021; Avron & Toledo, 2011), a tool from randomized numerical linear, to efficiently compute the trace of Hessian without explicitly forming the Hessian matrix. The details are summarized in Algorithm 1.

To further verify pruning parameters controlled by minimizing Hessian trace is capable to decrease the condition number of the model, we empirically investigate the evolution of the trace of Hessian and the condition number. Here a small BERT with 4 Transformer layers is applied to WikiText-2 for language modeling. As shown in Figure 5, Masked-BERT where we prune 50% heads at epoch-10 controlled by minimizing the trace of Hessian presents a small value of Hessian trace, accompanied with a smaller value of condition number. The phenomenon demonstrates that minimizing the Hessian trace is an effective way to control the condition number of the model.

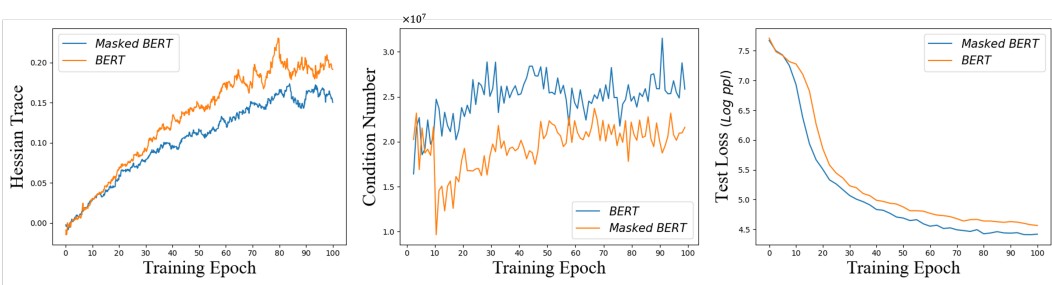

Figure 5: Connection between the trace of Hessian $Trace(H_f(\mathbf{m} * \mathbf{w}))$ and the condition number $L_f/\mu$.

---

**Algorithm 1** Hessian Trace Computation.

---

**Input:** Model Parameter: $\mathbf{m} * \mathbf{w}$, Model logits: $f$, Number of iteration: $n$
**Output:** Hessian trace of $f$: Trace
  1: Trace = 0
  2: **for** $i = 1, 2, \ldots, n$ **do**
  3:     Sample vector $a$ from a standard normal: $a \sim \mathcal{N}(0, \mathbf{I})$
  4:     Compute differential map of $f$ with respect to $\mathbf{w}$: $DF = \frac{\partial f}{\partial w}$.
  5:     Compute trace: Trace = Trace $+ a^\top \frac{\partial (DF^\top a)}{\partial w}$
  6: **end for**
  7: **return** Trace/n

---

**Algorithm 2** PL Regularization-Driven Pruning Algorithm.

---

**Input:** Training set $S$, pruning time and ratio $\{t_j, r_j\}_{j=1}^{q}$, maximum iterations $t_{max}$
  1: Initialize: $\mathbf{m} = \mathbf{1}$
  2: **while** not converged **do**
  3:     **for** $t = t_0, \ldots, t_{max}$ **do**
  4:         **if** $t = t_j$ **then**
  5:             Calculate $\mathbf{d}_{\mathbf{w}^{t_j}} = [\lambda_{min}(\mathcal{K}(\mathbf{w}_{[1]}^{t_j})), \rho(\mathcal{K}(\mathbf{w}_{[1]}^{t_j})), \ldots, \lambda_{min}(\mathcal{K}(\mathbf{w}_{[p]}^{t_j})), \rho(\mathcal{K}(\mathbf{w}_{[p]}^{t_j}))]$
  6:             **for** each batch in $S$ **do**
  7:                 $\mathbf{v} = \mathbf{v} - \eta \nabla_{\mathbf{v}} \mathcal{L}_{total}$ with $\alpha = 1$                     ▷ *Update* $\mathbf{v}$ *according to Eq. 36*
  8:             **end for**
  9:             $\mathbf{m}_{t_j} = TopK(g(\mathbf{d}_{\mathbf{w}}; \mathbf{v}), (1 - r_j)p)$                     ▷ *Generate binary mask* $\mathbf{m}$
 10:             $\mathbf{w}^{t_j} = \mathbf{m}^{t_j} * \mathbf{w}^{t_j}$                     ▷ *Reparameterize* $\mathbf{w}$
 11:         **end if**
 12:         **for** each batch in $S$ **do**
 13:             $\mathbf{w}^{t+1} = \mathbf{w}^t - \eta \nabla_{\mathbf{w}} \mathcal{L}_{total}$ with $\alpha = 0$                     ▷ *Update* $\mathbf{w}$ *according to Eq. 36*
 14:         **end for**
 15:     **end for**
 16: **end while**

---

### E.3 EXPERIMENTAL DETAILS OF BERT AND SWITCH-TRANSFORMER

BERT and Switch Transformer are trained from scratch by self-supervised Masked Language Model (MLM) task. The MLM objective is a cross-entropy loss on predicting the masked tokens. The models uniformly select 15% of the input tokens for possible replacement.

Considering the model capacity and computational resources, we train BERT on WikiText-2 and Switch Transformer on WikiText-103 datasets. The training hyperparameters are presented in Table 4. All baseline models, including our method, are trained by the same training setup. These experiments are conducted on $8 \times$ GPUs of NVIDIA GeForce RTX 3090.

We follow the original basic settings when training baseline models. For BERT-LTH and Switch-LTH, we use iterative magnitude pruning (IMP) (Chen et al., 2020) to target sparsity with rewinding step of 5% maximum training epochs. For Att-Score and Exp-Score, all the attention heads and experts across the model are sorted by gradient-based proxy importance score (Michel et al., 2019). Then they are pruned by the same iterative strategy as IMP. For SNIP and GraSP, we implement them in a structured way, i.e., we sum up the relative indicators in those components, and the attention heads and experts are pruned at the initialization stage of the training process (Lee et al., 2018; Wang et al., 2020). A protecting strategy is used for the pruning operation of all baselines and our model. We keep at least one head or expert in each of the layers to keep the activation across the model.

### E.4 EXPERIMENTAL DETAILS OF VGG-16

First, we summarize the architecture-specific pruning baselines, including FPGM, CHIP, and DPFPS. FPGM (He et al., 2019) compresses CNN models by pruning filters with norm- and distance-based criteria, which prunes all the weighted layers with the same pruning rate at the same

Table 4: Hyperparameters for training BERT and Switch Transformer

| Hyperparameter | BERT | Switch Transformer |
|---|---|---|
| Number of layers | 12 | 12 |
| Hidden size | 768 | 768 |
| Attention heads | 12 | 12 |
| Dropout | 0.1 | 0.1 |
| Sequence length | 512 | 512 |
| Batch size | 8 | 8 |
| Warmup steps | 0% | 6% |
| Weight decay | 0 | 1e-2 |
| Peak learning rate | 1e-4 | 2e-4 |
| Learning rate decay | Linear | Cosine |
| Adam $[\epsilon, \beta_1, \beta_2]$ | [0, 0, 0] | [1e-6, 0.9, 0.999] |
| Number of experts | | 4 |
| Capacity factor | | 1.5 |

time. CHIP (Sui et al., 2021) first extracts feature maps from a pre-trained model and calculates channel independence, which is used to sort and prune the filters, and then it fine-tunes the sparse pre-trained model. DPFPS (Ruan et al., 2021) prunes structured parameters in a dynamic sparsity manner, where the sparsity allocation ratios are distributed differently over layers in the training process.

Then, we report the detailed experimental settings. For training VGG-16 on CIFAR-10 and CIFAR-100, we use similar configurations as (Wang et al., 2020) does. For all baselines, the network is trained with Kaiming initialization (He et al., 2015) using SGD for 200 epochs. The learning rate is decayed from the initial value 0.1 by a factor of 0.1 at 1/2 and 3/4 of the total number of epochs. To obtain more stable results, we conduct each experiment in 3 trials.

In this paper, we evaluate the performance of all models on VGG-16 with 50% and 90% pruning ratios, respectively. The pruning policy of our method is the linear pruning schedule where starting from the epoch-15, we prune the same number of filters at each epoch until the target pruning ratio is reached. The pruning policy of baselines follows their origin settings given a target pruning ratio. In detail, for LTH, SNIP and GraSP, we conduct unstructured pruning with the target pruning ratio at the initialization stage. For FPGM, we utilize the norm-based criterion when given the 50% target pruning ratio. When given the 90% target pruning ratio, we first prune 50% filters by the norm-based criteria and then prune 40% filters by the distance-based criteria. For CHIP, given the 50% pruning ratio, following the original work, we assign the pruning ratio of the previous 7 layers as 20% and the ratio of remaining layers as 60%; while given the 90% pruning ratio, the ratio of the previous 7 layers is assigned as 75% and remaining layer as 95%.

## F  ADDITIONAL EXPERIMENTAL RESULTS

In this section, we report additional experimental results.

### F.1  RESULTS ON RESNET-56

This experiment focuses on ResNet-56 (Simonyan & Zisserman, 2014) trained on the CIFAR-10 dataset. The baselines include five filter pruning methods, namely FPGM (He et al., 2019), DPFPS (Ruan et al., 2021), LTH (Frankle & Carbin, 2018), SNIP (Lee et al., 2018), and GraSP (Wang et al., 2020). All models are trained from scratch and are pruned with a linear schedule, which starts from epoch-5 and prune the same number of filters per epoch until the target sparsity is reached.

Table 5 shows the performance of all methods trained on CIFAR-10 with a $25\%$ pruning ratio. As we can see, our method outperforms all pruning baselines in terms of training efficiency and test accuracy.

Table 5: Results of ResNet-56 on CIFAR-10 (Best results in Boldface).

| Method | CIFAR-10 | | | | |
| | Train Loss (@ epochs) | | Test Accuracy (@ epochs) | | |
| | @100 | @150 | @100 | @150 | Final |
|---|---|---|---|---|---|
| ResNet-56 | 0.1441 | 0.0148 | 91.76 | 92.80 | 93.43 |
| LHT | 0.1395 | 0.0178 | 92.18 | 92.98 | 93.50 |
| SNIP | 0.1509 | 0.0196 | 92.00 | 92.98 | 93.41 |
| FPGM | 0.1931 | 0.0689 | 91.27 | 92.60 | 93.41 |
| GraSP | 0.1453 | 0.0187 | 92.27 | 93.20 | 93.39 |
| DPFPS | 0.1507 | 0.0218 | 91.50 | 92.16 | 92.75 |
| Ours | **0.1380** | **0.0158** | **92.46** | **93.46** | **93.78** |

## F.2 TRAINING EFFICIENCY

This subsection shows the comparison of training efficiency by measuring the number of training iterations required for converging to a given test perplexity. We also evaluate the wall-clock time saved when the model converges to a given test perplexity. The calculation of parameter features and the training of the gating network are included in the statistic.

As shown in Figure 6, compared to vanilla BERT trained on WikiText-2, our model achieves the same test perplexity as the vanilla BERT at iterations 10k at iterations 8k, which is a $1.3\times$ speedup in terms of step time, i.e., our method only requires $0.7\times$ iterations to achieve the same test perplexity. As for the saved wall-clock time, as shown in Table 6, where the first row shows the wall-clock time in seconds and the second row shows the time saved compared with the vanilla BERT. We can observe that our method saves more wall-clock times compared with other baselines. In particular, it saves 29% wall-clock time of vanilla BERT. Furthermore, the training of the gating network accounts for only 2-3% of the total training time of BERT, demonstrating the efficiency of our method with the loop for pruning.

Table 6: Comparison of saved clock time on BERT.

| Methods | BERT | BERT-LTH | Att-Score | SNIP | GraSP | Ours |
|---|---|---|---|---|---|---|
| Wall-Clock Time (s) | 4112 | 4534 | 6683 | 3370 | 3563 | 2916 |
| Time Saved | 0% | -10% | -23% | 18% | 13% | 29% |

We find similar results in Switch-Transformer trained on WikiText-103. As shown in Figure 7 and Table 7, compared to vanilla Switch-Transformer, our method yields a 2x speedup, requiring 44% less clock time.

Table 7: Comparison of clock time saved when model achieves a given test perplexity. The symbol $*$ denotes expert pruning, and $**$ denotes expert and head pruning.

| Methods | BERT | Switch-LTH | Exp-Score | SNIP | GraSP | Ours* | Ours** |
|---|---|---|---|---|---|---|---|
| Time Saved (Approximately) | 0% | - | -9% | 19% | 8% | 32% | 44% |

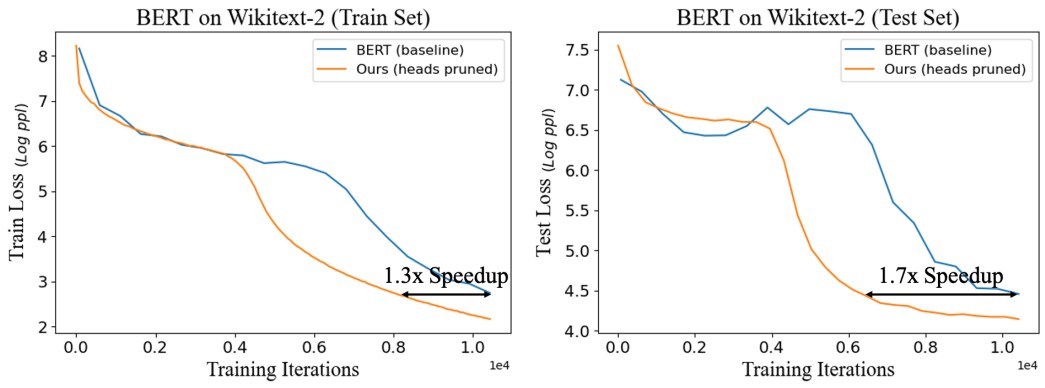

Figure 6: Training efficiency comparison of vanilla BERT and our model.

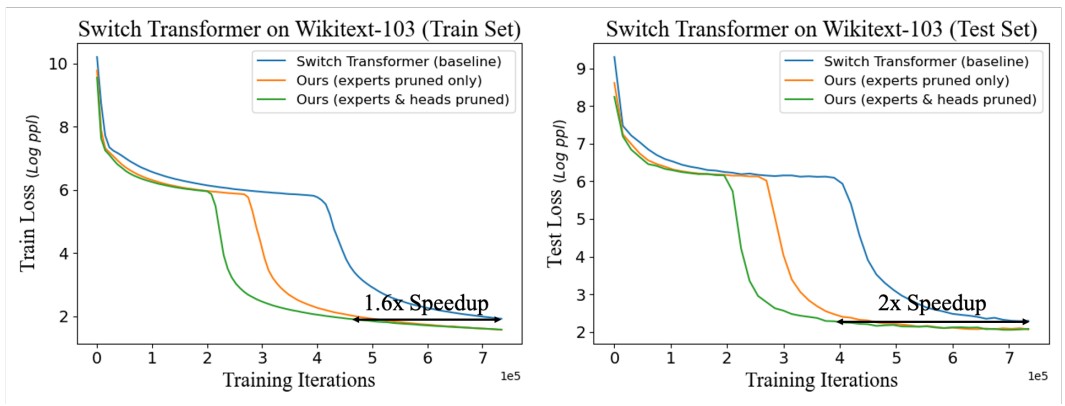

Figure 7: Training efficiency comparison of vanilla Switch-Transformer and our model.

### F.3 THE EFFECT OF PRUNING TIME

This subsection studies the influence of pruning time on the performance of our method on BERT. For a fixed target sparsity 75%, we compare the one-shot pruning schedule on epoch-0 (initialization), epoch-5, and epoch-50.

Table 8: Performance comparison on WikiText-2. For simplicity, we denote perplexity as PPL, and $\Delta$PPL represents the perplexity difference between training and test. Boldface indicates the best result.

| Method | Training PPL | | | Test PPL | $\Delta$PPL |
|--------|--------------|--------------|---------------|----------|-------------|
|        | @8 k iters   | @10 k iters  | @15 k iters   |          |             |
| [0]    | **13.26**    | 10.43        | **3.90**      | 67.49    | **6.27**    |
| [5]    | 15.72        | **9.63**     | 4.24          | **63.18**| 6.84        |
| [50]   | 37.86        | 15.4         | 5.63          | 87.18    | 7.19        |

As shown in Table 8, we empirically observe that it is sufficient to implement PL regularization on BERT in the early stages of training, i.e., important heads for BERT optimization can be discovered in the first 5 epochs.

We also evaluate the impact of one-shot pruning and linear pruning schedule for BERT. Specifically, we compare the performance of one-shot pruning and iterative pruning strategy on the heads of

Table 9: Comparsion of one-shot pruning and linear pruning schedule for BERT.

| Method | Training PPL | | | Test PPL | $\Delta$PPL |
|---|---|---|---|---|---|
| | @8 k iters | @10 k iters | @15 k iters | | |
| Bert(Baseline) | 39.37 | 13.49 | 6.35 | 75.57 | 0% |
| One-Shot | 15.72 | 9.63 | 4.24 | 63.18 | 29% |
| Iterative | 14.01 | 10.43 | 4.77 | 60.34 | 23% |

BERT with the same 75% pruning ratio. As shown in Table 9, the one-shot pruning strategy achieves competitive performance compared to the linear pruning schedule while saving more computational costs.

## F.4  THE EFFECT OF PRUNING RATIO

This subsection evaluates the performance of the proposed PL regularization-driven structured pruning method with varying pruning ratios.

Table 10 and Table 11 show the performance of BERT with the 50% and 90% pruning ratio, respectively.

Table 10: Performance Comparison of BERT on Wikitext-2. All models are pruned by 50% sparsity. Boldface indicates the best result among pruned models.

| Method | Training PPL | | | Test PPL | $\Delta$PPL |
|---|---|---|---|---|---|
| | @8 k iters | @10 k iters | @15 k iters | | |
| BERT | 39.37 | 13.49 | 6.35 | 75.57 | 7.49 |
| BERT-LTH | 21.98 | 12.48 | 4.41 | 74.44 | 8.48 |
| Att-Score | 19.09 | 12.81 | 4.75 | 73.55 | 7.28 |
| SNIP | 21.61 | 11.09 | 5.19 | 74.89 | 6.70 |
| GraSP | 19.47 | 10.82 | 4.34 | 78.34 | 8.71 |
| Ours | **13.82** | **7.82** | **3.51** | **67.49** | **6.49** |

Table 11: Performance Comparison of BERT on Wikitext-2. All models are pruned by 90% sparsity. Boldface indicates the best result among pruned models.

| Method | Training PPL | | | Test PPL | $\Delta$PPL |
|---|---|---|---|---|---|
| | @8 k iters | @10 k iters | @15 k iters | | |
| BERT | 39.37 | 13.49 | 6.35 | 75.57 | 7.49 |
| BERT-LTH | 75.64 | 14.86 | 6.32 | 81.78 | 6.74 |
| Att-Score | 33.05 | 17.08 | 8.16 | 84.61 | 5.88 |
| SNIP | 229.06 | 122.00 | 14.50 | 139.49 | 9.85 |
| GraSP | 91.38 | 22.44 | 8.32 | 94.25 | 8.00 |
| Ours | **24.24** | **11.27** | **5.46** | **72.68** | **5.45** |

Table 12 and Table 13 show the performance of VGG-16 with the 75% and 90% pruning ratio, respectively. We can observe that with a high pruning ratio, our method outperforms pruning baselines in terms of training efficiency and test performance.

Table 12: Performance Comparison of VGG-16 on CIFAR-10 and CIFAR-100. All models are pruned by 75% sparsity. Boldface indicates the best result among pruned models. Models without results mean that these models cannot converge under the current setting.

|  | Method & Dataset | VGG-16 | LHT | SNIP | FPGM | GraSP | CHIP | DPFPS | Ours |
|---|---|---|---|---|---|---|---|---|---|
| CIFAR-10 | Train Loss @ 100 epochs | 0.1090 | 0.1516 | 0.1046 | 0.2359 | 0.2220 | 0.3091 | **0.1037** | 0.1098 |
|  | Test Accuracy @ 100 epochs | 92.46 | 91.53 | 92.41 | 89.00 | 91.03 | 86.91 | 92.41 | **92.63** |
|  | Train Loss @ 150 epochs | 0.0075 | 0.0171 | 0.0095 | 0.0833 | 0.0470 | 0.1272 | 0.0090 | **0.0082** |
|  | Test Accuracy @ 150 epochs | 93.25 | 92.52 | 92.90 | 89.20 | 92.52 | 89.92 | 93.26 | **93.69** |
|  | Test Accuracy | 93.85 | 93.10 | 93.62 | 89.60 | 93.52 | 93.23 | 93.93 | **93.96** |
| CIFAR-100 | Train Loss @ 100 epochs | 0.4826 | **0.6250** | - | - | 1.108 | 1.2402 | 0.6399 | 0.6378 |
|  | Test Accuracy @ 100 epochs | 72.21 | **71.52** | - | - | 64.38 | 64.57 | 70.96 | 71.00 |
|  | Train Loss @ 150 epochs | 0.0459 | 0.0762 | - | - | 0.2601 | 0.6711 | 0.0755 | **0.0717** |
|  | Test Accuracy @ 150 epochs | 72.28 | 71.68 | - | - | 71.14 | 63.84 | 71.64 | **71.80** |
|  | Test Accuracy | 73.86 | 72.66 | - | - | 72.27 | 69.83 | 72.77 | **72.91** |

Table 13: Performance Comparison of VGG-16 on CIFAR-10 and CIFAR-100. All models are pruned by 90% sparsity. Boldface indicates the best result among pruned models. Models without results mean that these models cannot converge under the current setting.

|  | Method & Dataset | VGG-16 | LHT | SNIP | FPGM | GraSP | CHIP | DPFPS | Ours |
|---|---|---|---|---|---|---|---|---|---|
| CIFAR-10 | Train Loss @ 100 epochs | 0.1091 | 0.1867 | 0.1301 | - | 0.2530 | 0.3510 | 0.1417 | **0.1280** |
|  | Test Accuracy @ 100 epochs | 92.46 | 91.38 | 92.14 | - | 90.39 | 86.70 | 92.15 | **92.49** |
|  | Train Loss @ 150 epochs | 0.0075 | 0.0302 | 0.0132 | - | 0.0600 | 0.1871 | 0.0134 | **0.0122** |
|  | Test Accuracy @ 150 epochs | 93.25 | 91.83 | 93.07 | - | 92.40 | 89.02 | 92.97 | **93.11** |
|  | Test Accuracy | 93.85 | 93.19 | 93.56 | - | 93.18 | 93.18 | 93.59 | **93.61** |
| CIFAR-100 | Train Loss @ 100 epochs | 0.4826 | **0.8089** | - | - | 0.9990 | 1.7605 | 0.8767 | 0.8889 |
|  | Test Accuracy @ 100 epochs | 72.21 | 68.24 | - | - | 67.09 | 38.66 | 68.27 | **68.70** |
|  | Train Loss @ 150 epochs | 0.0459 | **0.1632** | - | - | 0.2460 | 1.2694 | 0.2050 | 0.2072 |
|  | Test Accuracy @ 150 epochs | 72.28 | 69.73 | - | - | 69.90 | 52.63 | 70.31 | **70.85** |
|  | Test Accuracy | 73.86 | 70.60 | - | - | 71.16 | 62.90 | 71.24 | **71.36** |

## F.5 FURTHER STUDY

Artificial deep learning models are typically over-parameterized, coming at the heavy computation effort during model training and inference. This work, drawing on theoretical insights, finds an efficient optimization strategy for the over-parameterized model to achieve a faster convergence time and a better test performance.

In practice, in order to find the optimal model, researchers usually train multiple models with varying parameter initializations. In over-parameterized settings, such a process requires too many computational resources and time efforts. Our method can not only reduce the computational cost for training one model, but also reduce the sensitivity to initialization and thus less models need to be trained.

To verify the latter benefit, we conduct experiments on two BERT models with different parameter initializations: BERT-A and BERT-B denote the BERT model with good initialization and bad initialization, respectively. Table 14 and Figure 8 show that imposing our method on the BERT-B can significantly improve its original performance; it outperforms the vanilla BERT-A and slightly underperforms the masked BERT-A model implemented with PL regularization. This phenomenon suggests the potential of our method in avoiding multiple initializations and meanwhile saving computation and time efforts.

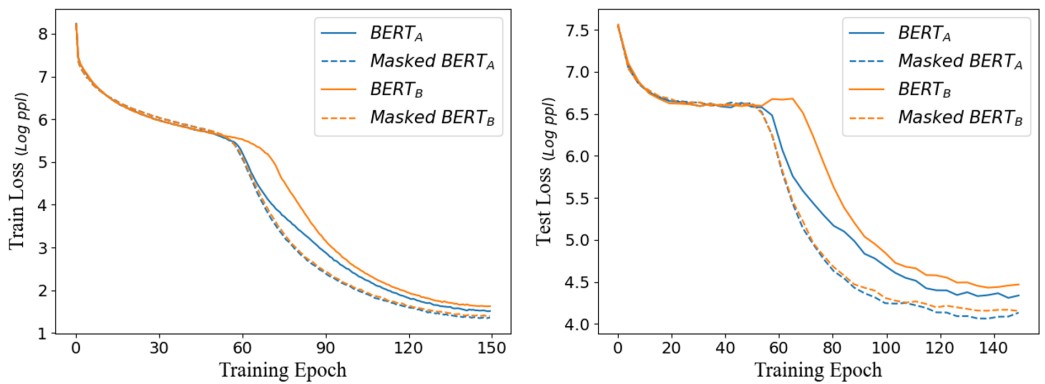

Figure 8: Training error and testing performance of BERT models with different parameter initialization.

Table 14: Results of BERT with different parameter initialization.

| Method | Training PPL | | | Test PPL | Wall-Clock |
| --- | --- | --- | --- | --- | --- |
| | @8 k iters | @10 k iters | @15 k iters | Final | Time Saved |
| BERT-A | 31.91 | 11.16 | 4.55 | 74.44 | |
| Masked BERT-A | 19.01 | 7.78 | 3.89 | 57.97 | 29.2% |
| BERT-B | 59.44 | 13.37 | 5.09 | 83.93 | |
| Masked BERT-B | 19.65 | 8.01 | 4.03 | 63.43 | 33.6% |

