# OpenReview forum: "Over-parameterized Model Optimization with Polyak-{\L}ojasiewicz Condition"
_ICLR.cc/2023/Conference — ICLR 2023 poster_

### Official Review · Reviewer_ng9v · 2022-10-15

**Confidence:** 4
**Correctness:** 3
**Technical Novelty And Significance:** 3
**Empirical Novelty And Significance:** 3
**Recommendation:** 6

**Clarity, Quality, Novelty And Reproducibility:**

Clarity: reasonable. The paper is easy to follow. But I have the feeling that the authors intentionally hide the transformation from condition number to trace estimation, which may delivery misleading signals to the readers.

Quality: I think this paper is an empirical paper, so it is difficult for me to evaluate the quality of the experimental results. It looks solid to me.

Novelty: I think it is interesting to use the trace of Hessian as the regularizer, to improve training and generalization.

Reproducibility: the experiments look reproducible to me.

**Strength And Weaknesses:**

This is a theory-inspired paper, so presumably the emprical ideas were originated from the theory community, instead of random hacking. It is hard for me to evaluate the experimental results as an expert, but it seems to me that all the results are pretty good, because they easily beat all the baselines.

However, it is worth pointing out that
1. The theory contribution of this paper is minimal. It seems to me that the main theory result is Theorem 6, which is a direct application of the definition of Pl condition and stability.
2. Regularizing the condition number is difficult. So what they do is convert it to the minimal eigenvalue of NTK. But even minimal eigenvalue of NTK is also difficult to compute, so they further convert it to the norm of Hessian. But the norm of Hessian is still expensive to compute, so they further convert it to the trace of Hessian. Finally, the trace of Hessian can be efficiently approximated using random vector sampling. After so many steps of transformations, it is hard to justify that the empirical success is indeed for the theory of condition number.

It seems to me that this paper can be greatly improved, if the authors can show that adding regularization of the trace of Hessian can help optimization & generalization, because it seems to be what the model is actually doing. The theory of NTK relies on pretty strong assumptions on the learning rate and network width, so it is difficult for me to believe that the condition number is the actual reason for the model to work.

**Summary Of The Paper:**

This is a theory-inspired empirical paper. The authors found that condition number (PL/Lipschitz) is an important parameter related to convergence and generalization, so they try to add condition number as the regularization term during training, and also use it to do pruning. It turns out that by doing that, the algorithm runs faster in training and generalizes better, compared with many baseline algorithms.



**Summary Of The Review:**

Since I think this paper is an empirical paper, and the connection between the trace estimation and condition number is weak, I currently give a weak reject.

However, I am willing to change it to weak accept, if the authors, or other reviewers can convince me, that the regularization + pruning techniques along is a very good contribution to the community.

-----After rebuttal ----
After reading the rebuttal, I think:
1. I still agree with Reviewer a5MJ that this is not a good theory paper.
2. I think the authors demonstrated that their methods are useful emprically to the community.
3. Other two reviewers seem to like the paper.
4. The authors demonstrated that there exists some connections between trace estimation and condition number.

So I raise the score.

---

> ### Author Response · Authors · 2022-11-15
> **Response to Reviewer ng9v**
>
> > **Q1:** The connection between the trace estimation and condition number is weak.
>
> **A1:** Thanks for your comment. We added Proposition 8 to show that the condition number is upper bounded by the Hessian norm $\|H_f\|_2$, and we added Theorem 9 to show that pruning parameters governed by minimizing the Hessian norm can decrease the upper bound of the condition number. To briefly explain, based on the Taylor expansion around initialization $\mathbf{w}_0$, we show that the pruning operation controlled by minimizing the Hessian norm is able to decrease the upper bound of Lipschitz constant of the pruned network; and based on the matrix theory, we show that pruning parameter is able to increase the minimal eigenvalue of NTK. Proposition 8, Theorem 9 and their proofs have been included in our revision (Appendix C, D).
>
> Since directly computing the Hessian and optimizing the Hessian norm is difficult, we propose to optimize the upper bound of Hessian norm, which can be effectively approximated by using random vector sampling without explicitly computing the entire Hessian. Empirical results show that pruning parameters controlled by penalizing the trace of Hessian leads to a decrease in the condition number. The result is included in Figure 4 of Appendix E.
>
> > **Q2:** The contribution to the community.
>
> **A2:** Artificial deep learning models are typically over-parameterized, imposing heavy computational load during model training and inference. This work, drawing on theoretical insights, finds an efficient optimization strategy for the over-parameterized model to achieve faster convergence and better test performance.
>
> In practice, in order to find the optimal model, researchers usually train multiple models with varying parameter initializations. In over-parameterized settings, such a process imposes heavy computational burdens. Our method can reduce the computational cost for training one model, and reduce the sensitivity to initialization values, and thus require fewer models need to be trained. To verify the latter benefit, we conduct experiments on two BERT models with different parameter initializations: BERT-A and BERT-B denote the BERT model with good initialization and bad initialization, respectively. Table 1 shows that using our method on BERT-B can significantly improve its original performance; it outperforms the vanilla BERT-A and slightly underperforms the masked BERT-A model implemented with PL regularization. This phenomenon suggests the potential of our method in avoiding multiple initializations thus reducing computational load.
>
> | Method | Training PPL  @8k iters | Training PPL @10k iters | Training PPL @15k iters | Test PPL (Final) | Wall-Clock Time Saved |
> |---|---|---|---|---|---|
> | BERT-A | 31.91 | 11.16 | 4.55 | 74.44 |  |
> | Masked BERT-A | 19.01 | 7.78 | 3.89 | 57.97 | 29.2\% |
> | BERT-B | 59.44 | 13.37 | 5.09 | 83.93 |  |
> | Masked BERT-B | 19.65 | 8.01 | 4.03 | 63.43 | 33.6\% |
>
> *Table 1: Results of BERT with different parameter initialization. Boldface indicates the best result.*

---

> > ### Author Response · Authors · 2022-11-21
> > **Response to Reviewer ng9v (New Experiment Result)**
> >
> > We would like to report another study by expanding the original model BERT-Base (110M) to BERT-Large (340M), and the training data from WikiText-2 (12.91 MB) to BookCorpus dataset (4629.00 MB). The pruning policy is one-shot pruning, i.e., we prune 50\% heads at epoch-5 during training.
> >
> > Table 2 shows the wall-clock times of the vanilla BERT-Large and the pruned BERT-Large, where the first row shows the wall-clock time in minutes and the second row shows the time saved compared with the vanilla BERT-Large. Note that the training of the gating network is included in the statistics. As we can see, using the proposed method, the training time (wall clock time) is reduced by 30\%-50\%.
> >
> > | Method | Clock Time (min) @ Log ppl = 2.5 | Clock Time (min) @ Log ppl = 2 | Clock Time (min) @ Log ppl = 1.5 |
> > |---|---|---|---|
> > | BERT-Large | 1151 | 1499 | 1905 |
> > | Masked BERT-Large | 636 | 758 | 1298 |
> > | Time Saved | 44.7% | 49.4% | 31.9% |
> >
> > *Table 2: Results of BERT-Large on BookCorpus dataset.*

---

### Official Review · Reviewer_nR2e · 2022-10-15

**Confidence:** 5
**Clarity, Quality, Novelty And Reproducibility:** See the comments in the section of "s…
**Correctness:** 2
**Technical Novelty And Significance:** 4
**Empirical Novelty And Significance:** 3
**Recommendation:** 8

**Strength And Weaknesses:**

Strength:

The idea of regularizing the PL condition (hence, the condition number) is novel and interesting. It is widely believed that the condition number directly controls the convergence speed, directly regularizing it should be helpful in optimization speed.

It is a fascinating idea to learn a mask to select the “good” weights (or weight groups) such that the resulting pruned network has a low condition number. This method makes it feasible to regularize the PL condition. I have never seen such ideas in the literature, and I think it is a good addition to the machine learning community.

The experimental results on several commonly used networks show that the proposed method is effective in improving the network performance.

Weaknesses:

Several *important* aspects of the proposed method and its implementation are missing, which restrict the quality of this paper. Addressing them should improve the paper. I list them below.

> The connection between the regularization term (condition number) $L_f/\lambda_{min}$ and the trace of Hessian matrix $H_f$ is not clearly established. The authors refers to Prop 11 for the connection, however, there is no explicit relation between the two quantities presented in Prop 11. The only relation is $\lambda_{min}/L_f \ge \lambda_0/L_f-2\sqrt{n}R\epsilon$. However, on the right hand side: the first term $\lambda_0/L_f$ which is the major contribution of the two is not obviously related to the Hessian, noting that $\lambda_0$ is the smallest eigenvalue of the NTK at initialization; the second term $-2\sqrt{n}R\epsilon$, which is negative, should have a much smaller magnitude than the first term, especially when the width is large.

> The regularization term in Eq. (6) and (7) are evaluated over a large set $\mathcal{W}$, typically large enough to cover the optimization trajectory, and in principle the trace of Hessian should be also taken minimization over $\mathcal{W}$; however, the trace of Hessian is evaluated on a single point $w$, see Eq. (41). This is a mismatch, and there should be some theory to justify this.
>>Note that, in most practical cases (including the experiments of this paper), the network is not super wide, and the Hessian matrices may be significantly different from each other within $\mathcal{W}$. In these cases, a single point evaluation (in Eq.(41)) should not be enough.

> Many important implementation details are missing. It is necessary to present them in the paper.

>> How is the NTK computed for the neural networks: BERT, VGG-16 etc? As mentioned in Section 3.2.2, quantities of the NTK (e.g., $\lambda_{min}$ and entropy Eq. (8)) are needed as input for the gating network. As far as I know, the NTK for these networks is very hard to compute. It is important to present the computation in the paper.

>> How is the gradient $\nabla_v L_{prune}$ computed? Especially, what is the expression of this gradient? It seems that the derivative goes through many complicated functions, for example, TopK, trace of Hessian.

>> The most important of these is that: how are the parameter subsets $w_{[i]}$ partitioned for the neural networks, BERT, VGG-16, etc? Difference partition methods may largely affect the performance.

I am also a bit suspicious about Figure 1(a). Condition numbers usually take very large values, typically greater than 1k. Moreover, in theory it should never be less than 1. However, the condition numbers in Figure 1(a) are quite small, even less than 1. I am confused about it, and hope the authors can give some explanation.

The paper only compares the convergence based on the number of epochs. As the proposed algorithm has an extra training module (training the gating network), I would like to see a comparison based on the wall-clock time also.

Definition 1 is actually the definition of interpolation, not over-parameterization. Although the two concepts often co-exist in many cases, they are not the same thing. A definition of over-parameterization has to be related to the number of model parameters.

I’d also like to know:
> at which epochs are the VGG-16 pruned? (I might miss this details, but I did not find it).

> performance of the proposed method on ResNets (e.g., ResNet-32, ResNet-50) with CIFAR-10.


**Summary Of The Paper:**

This paper proposes adding a PL condition related regularization term onto the optimization loss function of over-parameterized deep neural networks. The intuition was that, by regularizing the inverse of the PL condition, the condition number was decreased, which leads to a faster convergence rate. This regularization was implemented by training a simple gating network that predicts a mask on the weights, so that after applying the mask the pruned network has a smaller condition number. Experiments on BERT, Switch-Transformer and VGG-16 were conducted, and shows that the proposed method results in better performance.

**Summary Of The Review:**

In summary, the paper proposes very novel and interesting ideas in regularizing PL (or condition number) to improve convergence speed, and in implementing the regularization. The weak point is that it lacks a lot of important implementation details and theoretical justifications of some claims. I tempororily put it as below threshold. But I would consider it as a good paper, as long as my concerns are addressed and more details are included.


-------------After author feedback----------------------------------------------------

The authors provided very detailed feedback, which clarified many of the implementation details that was missing in the original submission. Most of my initial concerns are addressed, except this one:

> " The regularization term in Eq. (6) and (7) are evaluated over a large set $\mathcal{W}$ ... however, the trace of Hessian is evaluated on a single point $w$, see Eq. (41). This is a mismatch, and there should be some theory to justify this."

In addition, the revision, especially Figure 3, provides a directly experimental evidence that the proposed method is successful in decreasing the condition number by prunning, which is align with the motivation of this work.

Given the above facts, I would like to raise my score accordingly.

---

> ### Author Response · Authors · 2022-11-15
> **Response to Reviewer nR2e (Part 1/3)**
>
> > **Q1:** The connection between the regularization term (condition number) $L_f/\lambda_{min}$ and the trace of Hessian matrix $H_f$ is not clearly established.
>
> **A1:** Thanks for your comment. We added Proposition 8 to show that the condition number is upper bounded by the Hessian norm $\|H_f\|_2$, and we added Theorem 9 to show that pruning parameters governed by minimizing the Hessian norm can decrease the upper bound of the condition number. To briefly explain, based on the Taylor expansion around initialization $\mathbf{w}_0$, we show that the pruning operation controlled by minimizing the Hessian norm is able to decrease the upper bound of the Lipschitz constant of the pruned network; and based on the matrix theory, we show that pruning parameters are able to increase the minimal eigenvalue of NTK. Proposition 8, Theorem 9 and their proofs have been included in our revision (Appendix C, D).
>
> Since directly computing Hessian and optimizing the Hessian norm is difficult, we propose to optimize the upper bound of the Hessian norm, which can be effectively approximated by using random vector sampling without explicitly computing the entire Hessian. Empirical results show that pruning parameters controlled by penalizing the trace of Hessian leads to a decrease in the condition number. The result is included in Figure 4 of Appendix E.
>
> > **Q2:** The regularization term in Eq. (6) and (7) are evaluated over a large set $\mathcal{W}$, typically large enough to cover the optimization trajectory, and in principle the trace of Hessian should be also taken minimization over $\mathcal{W}$; however, the trace of Hessian is evaluated on a single point $w$, see Eq. (41). This is a mismatch, and there should be some theory to justify this.
>
> **A2:** Thanks for your comment. We conduct experiments and show that evaluating the trace of Hessian on a small number of parameter points achieves competitive performance with a large number of parameter points over the optimization trajectory. Specifically, we evaluate the performance of VGG-16 on CIFAR-10 with varying pruning policies. VGG-16-large and VGG-16-small are both implemented with the linear pruning schedule starting from epoch 15 with a 50\% target pruning ratio. VGG-16-large prunes 1\% filters per epoch until the 50\% target pruning ratio is reached, consequently, VGG-16-large evaluates the trace of the Hessian on parameter points over $15$ to $65$ epochs. VGG-16-small prunes 2.5\% filters per epoch and evaluates the trace of Hessian over $15$ to $35$ epochs. Compared with VGG-16-small, VGG-16-large evaluates the trace of the Hessian on more parameter points over the optimization trajectory. As shown in the Table 1, VGG-16-small achieves competitive performance compared to VGG-16-large with much fewer computational costs.
>
> | Method | Over Epochs | Train Loss @100 epochs | Train Loss @150 epochs | Test Accuracy @100 epochs | Test Accuracy @150 epochs | Test Accuracy (Final) |
> |:---:|---|:---:|:---:|:---:|:---:|:---:|
> | VGG-16 | - | 0.1441 | 0.0148 | 91.76 | 92.89 | 93.43 |
> | VGG-16-Large | 15-65 | 0.1040 | **0.0068** | 92.70 | **93.55** | **94.27** |
> | VGG-16-Small | 15-35 | **0.1029** | 0.0076 | **92.91** | 93.50 | 94.07 |
>
> *Table 1: Results of VGG-16 on CIFAR-10 with evaluating the trace of Hessian on varying parameter point (Best results in Boldface).*
>
> > **Q3:** How is the NTK computed for the neural networks: BERT, VGG-16 etc? As mentioned in Section 3.2.2, quantities of the NTK (e.g., $\lambda_{min}$ and entropy Eq. (8)) are needed as input for the gating network. As far as I know, the NTK for these networks is very hard to compute. It is important to present the computation in the paper.
>
> **A3:** Thanks for your question. Quantities of the NTK ($\lambda_{min}$ and entropy Eq. (8)) refer to the parameter features of each partitioned parameter $\mathbf{w}_ {[i]}$ (not the entire network). The partitioned parameter $\mathbf{w}_ {[i]}$ is a group of parameters, e.g., the parameters of channel $i$ for the convolutional neural network. Following previous work [1,2], given a partitioned parameter $\mathbf{w}_ {[i]}$, each entry of the corresponding NTK matrix ($\mathcal{K}(\mathbf{w}_ {[i]})$) is the dot product of two gradient vectors, and thus NTK matrix ($\mathcal{K}(\mathbf{w}_ {[i]})$) is equivalent to the Gram matrix of per-sample gradients. In our experiments, we randomly sample 50 batches of data samples to obtain an approximated NTK matrix with a size of (50 $\cdot$ batch-size) $\times$ (50 $\cdot$ batch-size), which is acceptable for memory usage.

---

> > ### Author Response · Authors · 2022-11-15
> > **Response to Reviewer nR2e (Part 2/3)**
> >
> > > **Q4:** How is the gradient $\nabla_\textbf{v} L_{prune}$ computed? Especially, what is the expression of this gradient? It seems that the derivative goes through many complicated functions, for example, TopK, trace of Hessian.
> >
> > **A4:** Recall that the objective function is given by $\mathcal{L}_ {prune} =  \mathcal{L}_ S(\mathbf{m} \ast \mathbf{w}) + \alpha Trace(H_f( \mathbf{m} \ast \mathbf{w}))$. The gradient $\nabla_\textbf{v} \mathcal{L}_{prune}$ contains two parts of derivatives, including the gradient of empirical error $\nabla_\textbf{v} \mathcal{L}_S(\mathbf{m} * \textbf{w})$ and the gradient of Hessian trace $\nabla_\textbf{v} Trace(H_f( \mathbf{m} \ast \mathbf{w}))$.
> >
> > The first term is calculated as $\nabla_ \textbf{v} \mathcal{L}_ S(\mathbf{m} * \textbf{w}) = \nabla_ \textbf{v} \mathcal{L}_ S(g(d_\textbf{w};\textbf{v}) * \textbf{w})$, where $g(d_\textbf{w};\textbf{v})$ represents a differentiable function. As for the computation of second term, we first approximate the trace of Hessian using the random vector sampling method (summarized in Algorithm 2), and then calculate the gradient with respect to the parameter $\mathbf{v}$, represented as $\nabla_\textbf{v} Trace \left(H_f\left(g\left(\mathbf{d}_{\mathbf{w}} ; \mathbf{v}\right) \ast \mathbf{w}\right)\right)$.
> >
> > Note that the $TopK$ function is not involved when training the gating network but is used to generate a binary mask $\mathbf{m} = TopK (g,k)$ once the updates of parameter $\mathbf{v}$ is completed. We added a detailed pseudo-algorithm (Algorithm 2) to illustrate our method in Appendix E.2.
> >
> > > **Q5:** The most important of these is that: how are the parameter subsets $w_ {[i]}$ partitioned for the neural networks, BERT, VGG-16, etc? Difference partition methods may largely affect the performance.
> >
> > **A5:** Thanks for your question. We set the partition methods of different architectures according to their inherent model architecture.For BERT, the parameter subsets $\mathbf{w}_ {[i]}$ refer to the independent heads of the multi-head attention module. For Switch-transformer, the parameter subsets $\mathbf{w}_ {[i]}$ refer to the independent expert of the mixture-of-experts module.For VGG-16 and ResNet, the parameter subsets $\mathbf{w}_ {[i]}$ refer to the independent convolutional filters.
> >
> > > **Q6:** I am also a bit suspicious about Figure 1(a). Condition numbers usually take very large values, typically greater than 1k. Moreover, in theory it should never be less than 1. However, the condition numbers in Figure 1(a) are quite small, even less than 1. I am confused about it, and hope the authors can give some explanation.
> >
> > **A6:** Thanks for your comment. The condition numbers in Figure 1(a) are scaled by $10^{-8}$. We add the scaling factor in Figure 1(a) for the revised manuscript.
> >
> > > **Q7:** The paper only compares the convergence based on the number of epochs. As the proposed algorithm has an extra training module (training the gating network), I would like to see a comparison based on the wall-clock time also.
> >
> > **A7:** Thanks for your comment. As suggested, we measure the wall-clock time of vanilla BERT and pruned BERT using several pruning-based optimization algorithms; the results are listed in Table 2, where the first row shows the wall-clock time in seconds and the second row shows the time saved compared with the vanilla BERT. Note that the training of the gating network is included in the statistic.
> >
> > We can observe that our method saves more wall-clock times compared with other baselines. In particular, it requires 29\% less wall-clock time than vanilla BERT. Furthermore, the training of the gating network accounts for only 2-3\% of the total training time of BERT, demonstrating the efficiency of our method with the loop for pruning.
> >
> > | Methods | BERT | BERT-LTH | Att-Score | SNIP | GraSP | Ours |
> > |---|---|---|---|---|---|---|
> > | Wall-Clock Time (s) | 4112 | 4534 | 6683 | 3370 | 3563 | 2916 |
> > | Time Saved | 0\% | -10\% | -23\% | 18\% | 13\% | 29\% |
> >
> > *Table 2: Comparison of saved wall-clock time.*
> >
> > > **Q8:** Definition 1 is actually the definition of interpolation, not over-parameterization. Although the two concepts often co-exist in many cases, they are not the same thing. A definition of over-parameterization has to be related to the number of model parameters.
> >
> > **A8:** Thanks for your comment. We revise the definition of over-parameterization, i.e., a model is an over-parameterized model if the number of parameters $m$ is larger than the number of data.
> >
> > > **Q9:** at which epochs are the VGG-16 pruned? (I might miss this details, but I did not find it).
> >
> > **A9:** Thanks for your comment. The pruning strategy of VGG-16 is a linear pruning schedule which starts from epoch 15; every epoch prunes the same number of filters until the target sparsity is reached. We added the above implementation details in the revised version.

---

> > > ### Author Response · Authors · 2022-11-15
> > > **Response to Reviewer nR2e (Part 3/3)**
> > >
> > > > **Q10:** performance of the proposed method on ResNets (e.g., ResNet-32, ResNet-50) with CIFAR-10.
> > >
> > > **A10:** Thanks for your comment. Following your suggestion, we conduct experiments on ResNet-56 using the CIFAR-10 dataset with a 25\% pruning ratio. The pruning strategy is the linear pruning schedule, which starts from epoch 15 and prunes the same number of filters at every epoch until the target sparsity is reached. As shown in Table 3, our method outperforms five pruning baselines in terms of training efficiency and test accuracy on CIFAR-10. Especially, compared with the vanilla ResNet-56, our method achieves a higher accuracy with fewer parameters. This study has been included in Appendix F.1.
> > >
> > > | Method | Train Loss @100 epochs | Train Loss @150 epochs | Test Accuracy @100 epochs | Test Accuracy @100 epochs | Test Accuracy (Final) |
> > > |:---:|:---:|:---:|:---:|:---:|:---:|
> > > | ResNet-56 | 0.1441 | 0.0148 | 91.76 | 92.80 | 93.43 |
> > > | LHT | 0.1395 | 0.0178 | 92.18 | 92.98 | 93.50 |
> > > | SNIP | 0.1509 | 0.0196 | 92.00 | 92.98 | 93.41 |
> > > | FPGM | 0.1931 | 0.0689 | 91.27 | 92.60 | 93.41 |
> > > | GraSP | 0.1453 | 0.0187 | 92.27 | 93.20 | 93.39 |
> > > | DPFPS | 0.1507 | 0.0218 | 91.50 | 92.16 | 92.75 |
> > > | Ours | 0.1380 | **0.0158** | **92.46** | **93.46** | **93.78** |
> > >
> > > *Table 3: Results of ResNet-56 on CIFAR-10 (Best results in Boldface).*
> > >
> > > ---
> > > **References:**
> > >
> > > [1] Mok, Jisoo, et al. "Demystifying the Neural Tangent Kernel from a Practical Perspective: Can it be trusted for Neural Architecture Search without training?." Proceedings of the IEEE/CVF Conference on Computer Vision and Pattern Recognition. 2022.
> > >
> > > [2] Xu, Jingjing, et al. "KNAS: green neural architecture search." International Conference on Machine Learning. PMLR, 2021.

---

> > ### Comment · Reviewer_nR2e · 2022-11-17
> > **reply to authors**
> >
> > I thank the authors for the very detailed explaination, updating the theory and new experimental results.
> >
> > Given the non-rigorousness of some aspects of the theoretical argument, I think one direct (and somehow necessary) way to verify the theory is to numerically compare the condition numbers in the following two ways: 1) experimentally verify whether the condition number is decreasing during the training; 2) numerically verify whether the condition number of the pruned network is less than the condition number of the corresponding non-pruned network (the network trained without the regularization).
> >
> > I hope the authors still have time to verify this point before the deadline. I apologize for not raising this point earlier.

---

> > > ### Author Response · Authors · 2022-11-18
> > > **Response to Reviewer nR2e**
> > >
> > > Thanks for your reply. Following your suggestion, we empirically investigate the evolution of condition number of the pruned model and vanilla model to verify the claim of theories.
> > >
> > > We apply a small BERT with 4 Transformer layers to WikiText-2 for the language modeling task. We prune the heads of vanilla BERT by using a strategy of pruning 5\% heads every 10 epochs, controlled by minimizing the Hessian trace; the pruned network is denoted as Masked-BERT.  Empirical results show that compared with vanilla BERT, Masked-BERT presents a smaller value of condition number.
> > > Especially, we can observe a sharp decrease in the condition number of Masked-BERT after each pruning operation; to highlight this, red lines are included in the figure which indicate the condition number before and after pruning.
> > >
> > > The study and result are included in Appendix C.

---

### Official Review · Reviewer_a5MJ · 2022-10-22

**Confidence:** 4
**Correctness:** 2
**Technical Novelty And Significance:** 1
**Empirical Novelty And Significance:** 3
**Recommendation:** 6

**Clarity, Quality, Novelty And Reproducibility:**

There is much room for improvement in the quality and clarity of the paper. In particular, inconsistent notations as commented above should be fixed.

The proposed method itself is novel and the empirical performance is good.

Other comments on the clarity and quality:

- In the proof of sample complexity, the quadratic growth condition (27) between $w_1$ and $w_1^*$ is used, which is not true in general. Here, for condition (27) to hold, $w_1^*$ should be the projection of $w_1$ on the solution space.
- The Lipschitz continuity is used for the gradient $\nabla L_S(w)$ as well as $L_S(w)$ in the proof of exponential convergence, which should be clarified.
- Assumption in Theorem 10 seems strong. In particular, my concern is the convergence to an optimal solution regardless of the dataset $S$. It would be better to provide an example for this assumption. Moreover, the statement of Theorem 10 is ambiguous. What is $\epsilon_f$?

**Strength And Weaknesses:**

**Strengths**:

The idea of pruning the network to reduce the condition number is original and interesting. Moreover, the empirical performance seems good.

**Weaknesses**:

- My main concern is the correctness of the theory. Specifically, there is a conflict in assumptions. Specifically, PL-condition implies the quadratic growth condition as shown in [Karimi, et al, (2016)]. Hence, the objective function should be unbounded which contradicts the boundedness and Lipschitz continuity of the objective (assumed in Theorem 6). Some techniques to guarantee the boundedness of the parameters are probably needed.
- The notation is inconsistent and confusing. Both notations $L(w)$ and $L_S(w)$ are used for empirical risk, although the authors say "*we denote the loss function by L(x,y;w), or simply L(w), and also use L_s(w) to denote the empirical risk*". For instance, the notation $L(w)$ in Definitions 4 and 5 and Theorem 6 should be the empirical risk. There is also the description "*We denote $L$ and $L_S$ as the expected risk and empirical risk, respectively*" in Section B.2.1.

**Summary Of The Paper:**

This paper proposes a new method for pruning overparameterized neural networks. The method is based on the intuition that the smaller condition number leads to better optimization and generalization. That is, the proposed method is designed so that the pruned neural network has a small condition number. The experiments verify the effectiveness of the method on training large-scale networks such as BERT, Switch-Transformer, and VGG-16.

**Summary Of The Review:**

Although the empirical performance of the proposed method is good, there are several concerns about the correctness, quality, and clarity.

---

> ### Author Response · Authors · 2022-11-15
> **Response to Reviewer a5MJ (Part 1/2)**
>
> > **Q1:** My main concern is the correctness of the theory. Specifically, there is a conflict in assumptions. Specifically, PL-condition implies the quadratic growth condition as shown in [Karimi, et al, (2016)]. Hence, the objective function should be unbounded which contradicts the boundedness and Lipschitz continuity of the objective (assumed in Theorem 6). Some techniques to guarantee the boundedness of the parameters are probably needed.
>
> **A1:** Thank you for expressing your concern. This work focuses on over-parameterized models where the number of parameters is greatly larger than the data points. For example, wide residual networks use more parameters than the training samples. It is widely believed that, with appropriate parameters, an over-parameterized network could fit the data exactly. A more rigorous mathematical framework to analyze the theoretical properties of over-parameterization is the PL$^*$ condition. As shown in recent work~[1], the PL$^*$ condition ensures the existence of optimal solutions and that convergence holds in a ball of sufficient radius for the over-parameterized models.
>
> The PL$^*$ condition, $\|\nabla \mathcal{L}(\mathbf{w}) \|^2 \geq \mu \cdot \mathcal{L}(\mathbf{w})$, requires that gradient grow faster than a quadratic function as the model moves away from the optimal function value, implying the objective function has an upper bound. As discussed in [2], the quadratic growth~(QG) condition is a weaker condition than the PL condition. Although the QG condition only gives a lower bound on the objective function, it does not mean that the objective function should be unbounded. Specifically, following previous works [1, 3-5], we assume the PL condition (also the QG condition) holds in a ball around the initialization with a certain radius, which indicates that the loss function in Theorem 6 is not unbounded.
>
> > **Q2:** The notation is inconsistent and confusing. Both notations $\mathcal{L}$ and $\mathcal{L}_S$ are used for empirical risk (in definition 4, definition 5 and theorem 6). There is also the description ``We denote $\mathcal{L}$ and $\mathcal{L_S}$ as the expected risk and empirical risk, respectively" in Section B.2.1.
>
> **A2:** Thanks for your comment. We have carefully checked the notations again and made the following changes, as well as some clarifications.
>
> Definitions 4 and 5 are the definitions of Lipschitz continuity and PL condition, applicable to any general function. Thus, we revised the notations from $\mathcal{L}$ to $f$ to indicate its generality.
>
> In Theorem 6, we assume that the loss function satisfies Lipschitz continuity and the PL condition and, therefore, the notation should remain as $\mathcal{L}(\boldsymbol w)$.
>
> In Appendix B.2.1, $\mathcal{L}$ denotes the loss function and $\mathcal{L}_S$ denotes the empirical loss evaluated on the dataset $S$. We revised the notations of $\mathcal{L}$ and $\mathcal{L}_S$ in Appendix B.2.
>
> > **Q3:** In the proof of sample complexity, the quadratic growth condition (27) between $\mathbf{w}_1$ and $\mathbf{w}_1^*$ is used, which is not true in general. Here, for condition (27) to hold, $\mathbf{w}_1^*$ should be the projection of $\mathbf{w}_1$ on the solution space.
>
> **A3:** This work focuses on the optimization of over-parameterized models which typically satisfy the PL$^*$ condition on most of the parameter space. As discussed in [2], the PL condition implies the QG condition. Thus Equation (27) holds in the over-parameterized settings. Our work follows prior work [6], which studied the stability of the model satisfying the PL condition and QG condition. Building on it, we consider the PL$^*$ condition as it characterizes over-parameterized models more appropriately, and consequently, we provide a sharper bound of sample complexity of the model based on the PL$^*$ condition.
>
> > **Q4:** The Lipschitz continuity is used for the gradient $\nabla L_S$ as well as $\partial L$ in the proof of exponential convergence, which should be clarified.
>
> **A4:** Thanks for your comment. We proofread the assumption of Theorem 6 in the revised manuscript, i.e., the gradient of the loss function should also be $L_f$-Lipschitz continuous.

---

> > ### Author Response · Authors · 2022-11-15
> > **Response to Reviewer a5MJ (Part 2/2)**
> >
> >
> > > **Q5:** Assumption in Theorem 10 seems strong. In particular, my concern is the convergence to an optimal solution regardless of the dataset $\mathcal{S}$.
> > It would be better to provide an example for this assumption.
> > Moreover, the statement of Theorem 10 is ambiguous. What is $\epsilon_f$?
> >
> > **A5:** Thanks for your comment. Over-parameterization typically means that the number of parameters is larger than the number of data points, enabling the convergence to optimal solutions. For a more rigorous analysis on the convergence of over-parameterized models, we would refer readers to [1, 7-10].
> >
> > We revised the statement of Theorem 10 where $\epsilon_f$ is defined as a constant denoting the bound of empirical error gap between the optimal solution $\mathbf{w}^*$ and the obtained solution $\mathbf{w}$, i.e.,  $\left|\mathcal{L}_S\left(\mathbf{w}_S\right)-\mathcal{L}_S\left(\mathbf{w}_S^*\right)\right| \leq \epsilon_f$.
> >
> > ---
> > **References:**
> >
> > [1] Liu, Chaoyue, Libin Zhu, and Mikhail Belkin. "Loss landscapes and optimization in over-parameterized non-linear systems and neural networks." Applied and Computational Harmonic Analysis 59 (2022): 85-116.
> >
> > [2] Karimi, Hamed, Julie Nutini, and Mark Schmidt. "Linear convergence of gradient and proximal-gradient methods under the polyak-łojasiewicz condition." Joint European conference on machine learning and knowledge discovery in databases. Springer, Cham, 2016.
> >
> > [3] Liu, Chaoyue, Libin Zhu, and Mikhail Belkin. "Toward a theory of optimization for over-parameterized systems of non-linear equations: the lessons of deep learning." arXiv preprint arXiv:2003.00307 (2020).
> >
> > [4] Goujaud, Baptiste, Adrien Taylor, and Aymeric Dieuleveut. "Optimal first-order methods for convex functions with a quadratic upper bound." arXiv preprint arXiv:2205.15033 (2022).
> >
> > [5] Polyak, Boris Teodorovich. "Gradient methods for minimizing functionals." Zhurnal Vychislitel'noi Matematiki i Matematicheskoi Fiziki 3.4 (1963): 643-653.
> >
> > [6] Charles, Zachary, and Dimitris Papailiopoulos. "Stability and generalization of learning algorithms that converge to global optima." International Conference on Machine Learning. PMLR, 2018.
> >
> > [7] Du, Simon S., et al. "Gradient descent provably optimizes over-parameterized neural networks." arXiv preprint arXiv:1810.02054 (2018).
> >
> > [8] Bottou, Léon, Frank E. Curtis, and Jorge Nocedal. "Optimization methods for large-scale machine learning." Siam Review 60.2 (2018): 223-311.
> >
> > [9] Lei, Yunwen, and Ke Tang. "Learning rates for stochastic gradient descent with nonconvex objectives." IEEE Transactions on Pattern Analysis and Machine Intelligence 43.12 (2021): 4505-4511.
> >
> > [10] Belkin, Mikhail. "Fit without fear: remarkable mathematical phenomena of deep learning through the prism of interpolation." Acta Numerica 30 (2021): 203-248.

---

> > > ### Comment · Reviewer_a5MJ · 2022-12-01
> > > **Thanks**
> > >
> > > Thanks for your answers and your explanations. My concerns were well addressed and the corresponding revisions make the paper mathematically rigorous. In particular, an additional ball constraint in Theorem 6 resolves my concern about the objective function. (However, the current statement of Theorem 6 allows R to be arbitrarily small. A condition on the radius R may be needed so that a ball can cover a sufficient region.)
> > >
> > > Moreover, I would like to acknowledge the experimental contribution is significant. Thus, I would like to increase the score.

---

### Official Review · Reviewer_QD8B · 2022-10-28

**Confidence:** 3
**Correctness:** 4
**Technical Novelty And Significance:** 3
**Empirical Novelty And Significance:** Not applicable
**Recommendation:** 8

**Clarity, Quality, Novelty And Reproducibility:**

The paper is well rewritten, and proposes novel algorithms and achieves better empirical results. The reproducibility issue is addressed in the above "Strength And Weaknesses" section.

**Strength And Weaknesses:**

Strength:
- The motivation of the algorithm is well supported by a rigorous theoretical result.
- The empirical results demonstrate the algorithm works quite well.

Experimental weakness:
- The paper shows the results of training until convergence for VGG-16 but not for BERT or Switch-Transformer.
- The paper uses VGG-16 instead of more popular models in computer vision for the third experiment. Why?
- Is it possible to refer some other papers (better if official) where the results of training BERT on WikiText-2 and other experiments are report?
- Can we conclude the algorithm is more efficient than traditional optimization algorithms with the loop for pruning? It involves additional complexity.

Question:
- What will happen if we decrease the number of parameters in the model (such as BERT) gradually by tuning hidden dimension, for example? Does it achieve better performance as well.

**Summary Of The Paper:**

The paper proposed a novel method for regularizing over-parametrized models during optimization. The method is motivated by a theoretical result based on the condition number of the model. It improves the generalization performance empirically.

**Summary Of The Review:**

The paper proposes a novel algorithm based on a solid theoretical result. The algorithm achieves better generalization than baselines. However, there are several questions to be addressed in the experiment section.

---

> ### Author Response · Authors · 2022-11-15
> **Response to Reviewer QD8B**
>
> > **Q1:** The paper shows the results of training until convergence for VGG-16 but not for BERT or Switch-Transformer.
>
> **A1:** Thanks for your comment. The submitted version of the paper did show the test performance for BERT and Switch-Transformer until convergence. We revised the manuscript using the clearer notations.
>
> > **Q2:** The paper uses VGG-16 instead of more popular models in computer vision for the third experiment. Why?
>
> **A2:** Thanks for your question. The main reason is that VGG-16 is a typical over-parameterized model with 138M parameters. To further evaluate our method, we conduct experiments on ResNet-56 using the CIFAR-10 dataset with a target pruning ratio of 25\% and a linear pruning schedule, which starts from epoch 15 and prunes the same number of filters at each epoch until the target sparsity is reached. As shown in Table 1, our method outperforms five pruning baselines in terms of training efficiency and test accuracy on CIFAR-10. Especially, compared with the vanilla ResNet-56, our method achieves a higher accuracy with fewer parameters. This study has been included in Appendix F.1.
>
> | Method | Train Loss @100 epochs | Train Loss @150 epochs | Test Accuracy @100 epochs | Test Accuracy @100 epochs | Test Accuracy (Final) |
> |:---:|:---:|:---:|:---:|:---:|:---:|
> | ResNet-56 | 0.1441 | 0.0148 | 91.76 | 92.80 | 93.43 |
> | LHT | 0.1395 | 0.0178 | 92.18 | 92.98 | 93.50 |
> | SNIP | 0.1509 | 0.0196 | 92.00 | 92.98 | 93.41 |
> | FPGM | 0.1931 | 0.0689 | 91.27 | 92.60 | 93.41 |
> | GraSP | 0.1453 | 0.0187 | 92.27 | 93.20 | 93.39 |
> | DPFPS | 0.1507 | 0.0218 | 91.50 | 92.16 | 92.75 |
> | Ours | 0.1380 | **0.0158** | **92.46** | **93.46** | **93.78** |
>
> *Table 1: Results of ResNet-56 on CIFAR-10 (Best results in Boldface).*
>
> > **Q3:** Is it possible to refer some other papers (better if official) where the results of training BERT on WikiText-2 and other experiments are report?
>
> **A3:** Thanks for your suggestion. We added corresponding references for the results of training BERT[1], VGG-16[2] and ResNet-56[3] in the revised submission.
>
> > **Q4:** Can we conclude the algorithm is more efficient than traditional optimization algorithms with the loop for pruning? It involves additional complexity.
>
> **A4:** Thanks for your comment. Compared with the pruning-based optimization methods, our method is more efficient with the loop for pruning. Table 2 shows the wall-clock times of BERT using several pruning-based optimization algorithms and the proposed algorithm. Note that the training of the gating network is included in the statistic. We can observe that our method saves more wall-clock times compared with other baselines.  Furthermore, the training of the gating network accounts for only 2-3\% of the total training time of BERT, demonstrating the efficiency of our method with the loop for pruning.
>
> | Methods | BERT | BERT-LTH | Att-Score | SNIP | GraSP | Ours |
> |---|---|---|---|---|---|---|
> | Wall-Clock Time (s) | 4112 | 4534 | 6683 | 3370 | 3563 | 2916 |
> | Time Saved | 0\% | -10\% | -23\% | 18\% | 13\% | 29\% |
>
> *Table 2: Comparison of saved wall-clock time.*
>
> > **Q5:** What will happen if we decrease the number of parameters in the model (such as BERT) gradually by tuning hidden dimension, for example? Does it achieve better performance as well.
>
> **A5:** Thanks for your question. Compared with the one-shot pruning strategy currently used in the manuscript, gradually pruning the parameters of BERT slightly improves the performance at a cost of additional computational costs. In detail, we compare the performance of one-shot pruning and iterative pruning strategy on the heads of BERT with the same 75\% pruning ratio. As shown in Table 3, the one-shot pruning strategy achieves competitive performance compared to the iterative pruning strategy, while saving more computational cost. The study has been included in Appendix F.5.
>
> | Method | BERT (Baseline) | One-Shot | Iterative |
> |---|---|---|---|
> | Training Perplexity @ 8k updates | 39.37 | 15.72 | 14.01 |
> | Training Perplexity @ 10k updates | 13.49 | 9.63 | 10.43 |
> | Training Perplexity @ 15k updates | 6.35 | 4.24 | 4.77 |
> | Test Perplexity | 75.57 | 63.18 | 60.34 |
> | Wall-Clock Time Saved | 0\% | 29\% | 23\% |
>
> *Table 3: Comparison of different pruning strategy.*
>
> ---
> **References:**
>
> [1] Wang, Chenguang, Mu Li, and Alexander J. Smola. "Language models with transformers." arXiv preprint arXiv:1904.09408 (2019).
>
> [2] Li, Hao, et al. "Pruning filters for efficient convnets." arXiv preprint arXiv:1608.08710 (2016).
>
> [3] He, Kaiming, et al. "Deep residual learning for image recognition." Proceedings of the IEEE conference on computer vision and pattern recognition. 2016.

---

### Author Response · Authors · 2022-11-15
**General Response to Reviewers**

Thank you very much for reviewing our manuscript and providing detailed and constructive comments, which have been very helpful for us to improve the quality of our work. Enclosed, please see our answers to address the comments of individual reviewers.

Specifically, We have revised the manuscript to establish the connection between the regularization term (condition number) and the Hessian norm via Proposition 8 and Theorem 9. Furthermore, we add performance evaluation of the proposed method on ResNet-56 and we experiments on two BERT models with different parameter initializations to show the potential of our method to avoid multiple initializations and meanwhile saving computation and time efforts. Unless otherwise specified, section, page, and line numbers correspond to those in the revised manuscript.

---

### Decision · Program_Chairs · 2023-01-20

**Decision:**

Accept: poster

**Justification For Why Not Higher Score:**

The reviewers thought the theory was a bit incremental.

**Justification For Why Not Lower Score:**

All reviewers recommend acceptance.

**Metareview: Summary, Strengths And Weaknesses:**

This paper proposes a new method for pruning overparameterized neural networks. The method is based on the intuition that the smaller condition number leads to better optimization and generalization. That is, the proposed method is designed so that the pruned neural network has a small condition number. The experiments verify the effectiveness of the method on training large-scale networks such as BERT, Switch-Transformer, and VGG-16.

All reviewers agreed that this paper makes an interesting contribution to ICLR. They had some initial concerns about the theory, the writing and the experiments which were all addressed in the revision, and they have upgraded their score accordingly to all recommend acceptance.

**Note From Pc:**

if the above contains the word "oral" or "spotlight" please see: "oral" presentation means -> notable-top-5% and "spotlight" means -> notable-top-25%. As stated in our emails, we are disassociating presentation type from AC recommendations